# FROM PIXELS TO SEMANTICS: UNIFIED FACIAL ACTION REPRESENTATION LEARNING FOR MICRO-EXPRESSION ANALYSIS

**Yicheng Deng, Hideaki Hayashi, Hajime Nagahara**
The University of Osaka

## ABSTRACT

Micro-expression recognition (MER) is highly challenging due to the subtle and rapid facial muscle movements and the scarcity of annotated data. Existing methods typically rely on pixel-level motion descriptors such as optical flow and frame difference, which tend to be sensitive to identity and lack generalization. In this work, we propose D-FACE, a Discrete Facial ACtion Encoding framework that leverages large-scale facial video data to pretrain an identity- and domain-invariant facial action tokenizer, for MER. For the first time, MER is shifted from relying on pixel-level motion descriptors to leveraging semantic-level facial action tokens, providing compact and generalizable representations of facial dynamics. Empirical analyses reveal that these tokens exhibit order-dependent semantics, motivating sequential modeling. Building on this insight, we employ a Transformer with sparse attention pooling to selectively capture discriminative action cues. Furthermore, to explicitly bridge action tokens with human-understandable emotions, we introduce an emotion-description-guided CLIP (EDCLIP) alignment. EDCLIP leverages textual prompts as semantic anchors for representation learning, while enforcing that the "others" category, which lacks corresponding prompts due to its ambiguity, remains distant from all anchor prompts. Extensive experiments on multiple datasets demonstrate that our method achieves not only state-of-the-art recognition accuracy but also high-quality cross-identity and even cross-domain micro-expression generation, suggesting a paradigm shift from pixel-level to generalizable semantic-level facial motion analysis. Code is available on GitHub.

## 1 INTRODUCTION

Micro-expression recognition (MER) is a highly challenging task in affective computing because micro-expressions (MEs) are generally subtle (Scherer, 2005), rapid (less than 0.5 seconds) (Yan et al., 2013), and performed on local facial areas (Josephs, 2005). Nevertheless, due to the involuntary nature of MEs, their analysis is valuable in various applications that require uncovering hidden emotions, including lie detection (Ekman, 2009), medical care (Endres & Laidlaw, 2009), and national security (O'sullivan et al., 2009).

With the rapid development of deep learning, implementing accurate MER systems has become feasible. Early MER methods predominantly relied on hand-crafted pixel-level motion representations, such as optical flow (Gan et al., 2019) and frame difference (Li et al., 2022), to capture the fine-grained facial muscle movements. More recently, some researchers have explored end-to-end frameworks by employing autoencoders to estimate optical flow and then designing unique networks for recognition (Zhai et al., 2023). Nevertheless, hand-crafted optical flow remains the mainstream approach (Zhu et al., 2025).

While these pixel-level motion descriptors can describe subtle and local facial motion, they suffer from two major limitations. First, they are highly sensitive to identity and therefore lack cross-subject transferability, as shown in Figure 1(a). Second, they mainly describe displacement fields rather than the semantic meaning of facial muscle movements, which are more relevant to expression interpretation. In addition, the scarcity of annotated ME data further limits the robustness and generalization of deep learning models.

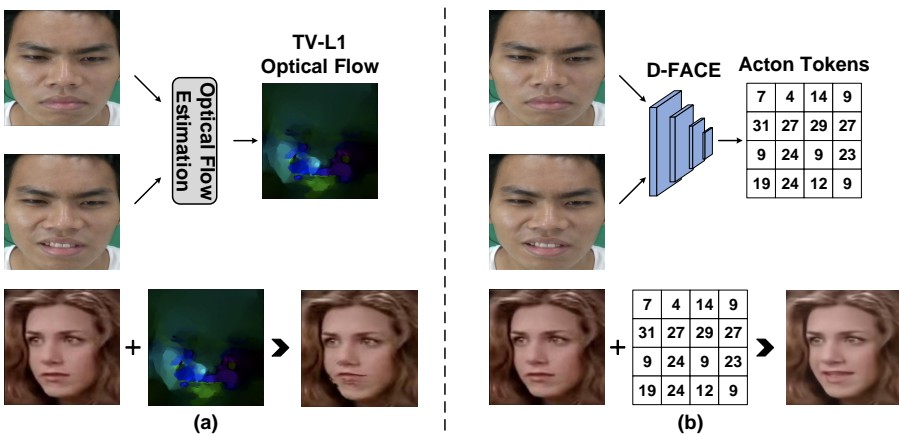

Figure 1: Motivation illustration. (a) Pixel-level motion descriptors (classical approaches) are identity-sensitive and suffer from poor generalization; (b) D-FACE encodes facial motions into action tokens, which are semantic motion representations that generalize across different identities and domains.

In this work, we aim to extract high-level semantic motion features that capture *what facial action occurs between two facial images*, instead of relying on pixel-level facial motion features. We expect that such semantic representations are inherently generalizable enough to be shared across different identities, datasets, and even domains. Moreover, they offer the potential to generate new ME data.

To this end, we propose D-FACE, a Discrete Facial ACtion Encoding framework pretrained on large-scale facial video data, which uses the objective of the vector-quantized variational autoencoder (VQ-VAE) (Van Den Oord et al., 2017) to discretize facial muscle movements between a pair of facial images into identity- and domain-invariant facial action tokens for MER. That is, the facial action tokens are generalizable to be shared across different identities and domains, thus serving as a unified vocabulary for representing facial actions. Through empirical studies, we observe that the facial action tokens exhibit order-dependent semantics and can be organized as one-dimensional (1D) sequences, similar to sentences that describe the facial muscle movements. Motivated by this observation, we employ a Transformer with sparse attention pooling to emphasize informative local cues for MER. Furthermore, we introduce an emotion-description-guided CLIP (EDCLIP) alignment to align the learned action tokens with human-understandable emotions. Specifically, our textual prompts are constructed by combining each emotion name with its typical facial action descriptions, providing a shared and consistent textual embedding for all samples of the same emotion and enabling a more explicit bridge between motion representations and emotional semantics. In addition, MER uniquely includes an ambiguous "others" category that cannot be precisely described through language. To address this, we adapt the CLIP mechanism by enforcing that "others" samples remain semantically distant from all emotion-specific textual embeddings rather than being aligned to an ill-defined description, allowing EDCLIP to better fit the characteristics of the MER task.

D-FACE achieves a paradigm shift for MER from observing pixel-level movements to modeling semantic-level action tokens. This is enabled by, for the first time, adapting VQ-VAE to MER through domain-specific considerations, carefully designed large-scale facial video pretraining, and extensive token–motion analysis that reveals interpretable and order-dependent token semantics. Extensive experiments demonstrate the effectiveness of D-FACE for MER and show that it learns a generalizable facial action representation, which extends beyond recognition and enables cross-subject and cross-domain ME generation.

Our contributions can be summarized as follows:

- We introduce D-FACE, a Discrete Facial ACtion Encoding framework pretrained on large-scale facial video data. For the first time, D-FACE provides a new perspective for MER by shifting from relying on pixel-level motion descriptors to using semantic-level facial action tokens that are identity- and domain-invariant.
- Through empirical analyses, we uncover that the learned facial action tokens exhibit order-dependent semantics, motivating us to model them as 1D sequences similar to sentences.

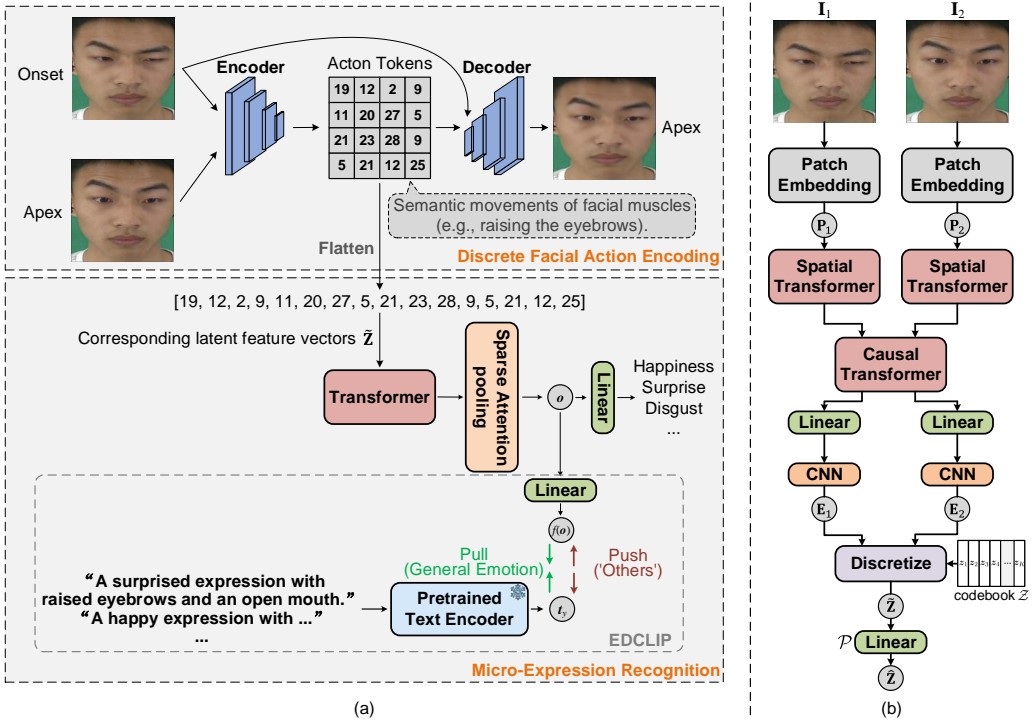

Figure 2: (a) Overview of the proposed framework. (b) Architecture of the action encoder. The action encoder discretizes facial muscle movements between two input facial images into facial action tokens in an unsupervised manner. These tokens are then flattened and processed by a Transformer with sparse attention pooling for MER. Finally, emotion-description-guided CLIP alignment bridges facial action tokens with human-understandable emotions.

Based on this insight, we employ a Transformer with sparse attention pooling to capture informative action cues for recognition.

- To explicitly bridge action tokens with human-understandable emotions, we introduce an emotion-description-guided CLIP alignment, where textual emotion anchors regularize representation learning. The "others" category, which lacks appropriate textual prompts due to its ambiguity, is pushed away from other textual embeddings.

- Extensive experiments on multiple benchmarks demonstrate state-of-the-art recognition performance and superior cross-subject and cross-domain generalization in ME generation.

## 2 RELATED WORKS

Over the past years, researchers have explored a variety of motion representations for MER. Early methods relied on hand-crafted features and traditional methods for MER. Zhao & Pietikainen (2007) introduced Local Binary Patterns from Three Orthogonal Planes (LBP-TOP) as a dynamic texture descriptor for MER. Wang et al. (2014) further proposed Local Binary Pattern Six Interception Points (LBP-SIP) for more compact representation.

With the development of deep learning, the combination of optical flow and neural networks has become the mainstream. OFF-ApexNet (Gan et al., 2019) computed optical flow between onset and apex frames and used CNNs for recognition. STSTNet (Liong et al., 2019) designed a triple-stream 3D CNN to capture multi-scale optical flow patterns. SLSTT (Zhang et al., 2022) combined short- and long-term optical flow with Transformer and LSTM for temporal modeling. SRMCL (Bao et al., 2024) introduced prototype-based contrastive learning with a self-expression reconstruction module to improve generalization. LTR3O (Zhu et al., 2025) proposed a reduced three-frame structure (onset, occurring, offset) and calibrated its expressiveness via learning-to-rank. More recently, autoencoder-

style optical flow estimation architectures (e.g., FlowNet (Dosovitskiy et al., 2015), RAFT (Teed & Deng, 2020)) have been adopted for MER under self-supervision (Zhai et al., 2023; Fan et al., 2023).

Beyond optical flow, alternative pixel-level motion descriptors have also been explored. LEAR-Net (Verma et al., 2019) compressed ME clips into a single dynamic image (Bilen et al., 2016) that contained motion information. Graph-TCN (Lei et al., 2020) applied video motion magnification to amplify subtle movements and extracted shape-based features. MMNet (Li et al., 2022) calculated frame differences between onset and apex frames as the motion feature and showed its effectiveness.

In contrast to these pixel-level motion descriptors, we propose to learn semantic-level facial action tokens, which are compact, identity- and domain-invariant, and thus enhance robustness and generalization for ME analysis.

## 3 METHOD

This section introduces our proposed D-FACE framework for MER. The overall architecture is illustrated in Figure 2(a). Given the onset and offset frames of an ME, we first discretize the facial muscle movements into a semantic facial action token map. These tokens are then flattened into a 1D sequence similar to sentences and modeled with a Transformer with sparse attention pooling, which emphasizes informative local cues for recognition. In addition, we align the learned facial action information with human-understandable emotions via emotion-description-guided CLIP alignment. We next describe each component in detail.

### 3.1 DISCRETE FACIAL ACTION ENCODING

The encoder architecture is illustrated in Figure 2(b). Given two temporally related facial images $\mathbf{I}_1$ and $\mathbf{I}_2$, our objective is to discretize their difference into identity- and domain-invariant action representations, which can be used to reconstruct $\mathbf{I}_2$ from $\mathbf{I}_1$ in an unsupervised manner. To this end, we adopt a conditional VQ-VAE (C-VQ-VAE) architecture originally introduced for latent action quantization in robotics (Ye et al., 2025). The original design in robotics employs a small codebook and a short latent action length suitable for discrete control, inspired by the latent action model in Genie (Bruce et al., 2024). These learned tokens are subsequently used as labels for a token-occurrence classification task for robotic control. In contrast, facial motion analysis requires modeling subtle and complex local muscle movements, which the original configuration cannot handle. Therefore, we adapt it to the facial domain by refining the codebook design and latent action length, and by conducting extensive token-motion investigations that reveal how tokens encode facial dynamics. Unlike general adaptation methods that introduce additional adapter networks (e.g., Chen et al. (2024)), our adaptation is achieved through a data- and analysis-driven redesign of the entire representation pipeline, enabling fine-grained facial motion extraction and understanding. We next describe the details of our framework.

Specifically, we first obtain the patch embeddings $\mathbf{P}_1$ and $\mathbf{P}_2$ via a linear projection. Each embedding is processed independently with a spatial Transformer to capture local spatial dependencies. The resulting features are subsequently passed through a causal Transformer to model the directed temporal transition from $\mathbf{I}_1$ to $\mathbf{I}_2$. A linear layer then reduces the dimensionality, and a CNN is employed to yield embedding maps $\mathbf{E}_1, \mathbf{E}_2 \in \mathbb{R}^{h \times w \times d'}$, where $h$ and $w$ denote the height and width of the embedding map, and $d'$ is the dimensionality of each embedding vector. The motion embedding is obtained as the difference between $\mathbf{E}_1$ and $\mathbf{E}_2$:

$$\mathbf{D} = \mathbf{E}_2 - \mathbf{E}_1. \tag{1}$$

We discretize $\mathbf{D}$ by mapping each feature vector to the nearest embedding from the codebook $\mathcal{Z} = \{\boldsymbol{z}_1, \ldots, \boldsymbol{z}_K\}$, producing an index map $M \in \mathbb{N}^{h \times w}$. The corresponding discretized action-token embeddings $\tilde{\mathbf{Z}} \in \mathbb{R}^{h \times w \times d'}$ are obtained by looking up the codebook:

$$(M)_{i,j} = \arg \min_{k \in \{1, \ldots, K\}} \|(\mathbf{D})_{i,j} - \boldsymbol{z}_k\|_2^2, \tag{2}$$

$$(\tilde{\mathbf{Z}})_{i,j} = \mathcal{Z}((M)_{i,j}), \tag{3}$$

where $i \in \{1, ..., h\}$, $j \in \{1, ..., w\}$. These action tokens capture the semantic movements of facial muscles between $\mathbf{I}_1$ and $\mathbf{I}_2$.

During training, instead of directly reconstructing $\mathbf{I}_2$ from $\tilde{\mathbf{Z}}$, NSVQ (Vali & Bäckström, 2022) is adopted before decoding:

$$\tilde{\mathbf{D}} = \mathbf{D} + \frac{\|\mathbf{D} - \tilde{\mathbf{Z}}\|}{\|\mathbf{V}\|}\mathbf{V}, \tag{4}$$

where $\mathbf{V} \in \mathbb{R}^{h \times w \times d'}$ is a tensor whose elements follow $\mathcal{N}(0,1)$. This encourages discretized facial action tokens to remain close to the real difference. The latent representation $\tilde{\mathbf{D}}$ is then up-projected back to the feature dimension through a linear projection layer $\mathcal{P}$ to obtain $\hat{\mathbf{D}}$:

$$\hat{\mathbf{D}} = \mathcal{P}(\tilde{\mathbf{D}}). \tag{5}$$

A spatial-Transformer-based decoder $\mathcal{D}$ takes $\mathbf{E}_1$ and $\hat{\mathbf{D}}$ as input to reconstruct the second frame $\hat{\mathbf{I}}_2$:

$$\hat{\mathbf{I}}_2 = \mathcal{D}(\mathrm{sg}[\mathbf{E}_1], \hat{\mathbf{D}}), \tag{6}$$

where $\mathrm{sg}[\cdot]$ denotes the stop-gradient operator.

Finally, the reconstruction loss minimizes the pixel-level discrepancy between the original and reconstructed second frame:

$$\mathcal{L}_{\mathrm{rec}} = \|\mathbf{I}_2 - \hat{\mathbf{I}}_2\|_2^2. \tag{7}$$

The quantization process in the C-VQ-VAE introduces a strong information bottleneck that forces the continuous motion features, obtained by differencing feature representations of diverse face pairs, to be mapped into a limited set of shared codebook vectors. Since our large-scale pretraining data contains a diverse set of facial identities with substantial variations in appearance and head pose (see Section 3.5 for pretraining details), a limited codebook cannot encode individual-specific morphological differences (e.g., how "opening mouth" manifests in a specific individual's mouth shape). These identity and appearance variations lie in a much larger and more complex subspace than the low-dimensional discrete codebook space. As a result, the optimization objective naturally encourages the quantization process to capture stable and commonly observed motion patterns shared across identities while filtering out identity- or appearance-specific variations, enabling the discretization to produce identity- and domain-invariant motion representations.

## 3.2 EMPIRICAL ANALYSIS OF ACTION TOKENS

Given a pair of facial images, we obtain an action token map from the C-VQ-VAE, which provides a compact and discrete representation of the facial muscle movements. To better understand how facial action tokens encode facial dynamics for subsequent MER, we conduct qualitative experiments and present the visualizations in Figure 3 (with more comprehensive analyses of token semantics provided in Appendix A.1 and A.2). We first input two identical facial images to the encoder to obtain an index map that corresponds to "no action" between frames. Then, we selectively alter a single index in the index map and generate a new facial image.

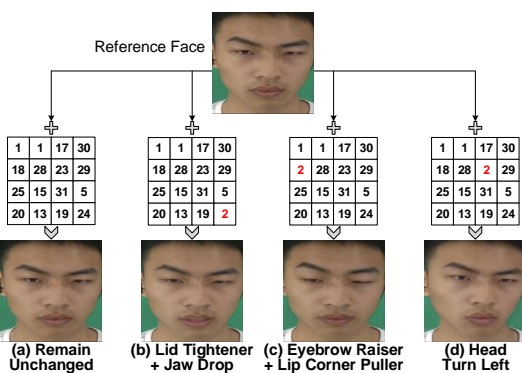

Figure 3: Visualization of the empirical analysis of action tokens through individual token alteration. The numbers in red stand for the altered indices.

These experiments yield two key observations. First, the same action token may activate different facial muscle motions when placed at different sequence locations, indicating that a token's semantics depend on its sequence location rather than being globally fixed. Second, tokens located in upper regions can also drive mouth-related actions in the lower facial area, while tokens in lower regions may affect eyebrow movements in the upper facial area. This indicates that tokens at a given 2D position do not activate facial actions in the corresponding facial region, but are instead determined by their relative ordering and contextual interactions. Unlike CNNs that rely on local geometric adjacency to organize spatial semantics, our tokenizer composes motion representations through global attention across all sequence locations

along with a globally shared codebook. Consequently, even global facial actions such as head movements can be controlled by manipulating a single token at a specific sequence location, rather than requiring the simultaneous activation of all tokens across the 2D grid (see Appendix A.2 for more detailed motion composition studies and visualization results). This motivates us to model the token map as a 1D ordered sequence using a Transformer with 1D positional embeddings. We next detail this design.

### 3.3 TRANSFORMER-BASED ACTION TOKEN MODELING

Motivated by our empirical observations, we flatten the discretized action-token embeddings $\tilde{\mathbf{Z}}$ as 1D sequences similar to sentences and then the linear projection layer $\mathcal{P}$ of the action encoder is used to obtain $\hat{Z} = (\hat{\boldsymbol{z}}_1, \hat{\boldsymbol{z}}_2, \ldots, \hat{\boldsymbol{z}}_L)$, where $L = h \times w$. To incorporate spatial information, we add a learnable positional embedding to each token. Formally, for the sequence $\hat{Z}$, their embedded representations are given by

$$\boldsymbol{g}_i = \mathcal{T}(\hat{\boldsymbol{z}}_i) + \boldsymbol{p}_i, \quad i = 1, \ldots, L, \tag{8}$$

where $\mathcal{T}(\cdot)$ is a token embedding layer and $\boldsymbol{p}_i \in \mathbb{R}^d$ is a learnable positional embedding.

The embedded sequence $\{\boldsymbol{g}_i\}_{i=1}^L$ is then processed by $N$ standard Transformer encoder layers (Vaswani et al., 2017), each consisting of multi-head self-attention and feed-forward sublayers, which enable context-aware interactions among facial action tokens.

In order to aggregate the sequence into a compact representation for classification, we introduce a sparse attention pooling mechanism. Specifically, given the output of the Transformer $X \in \mathbb{R}^{L \times d}$, we employ a global query vector $\boldsymbol{q} \in \mathbb{R}^d$ to calculate attention scores:

$$\boldsymbol{\alpha} = \text{softmax}\left(\frac{X\boldsymbol{q}}{\sqrt{d}}\right), \tag{9}$$

where $\boldsymbol{\alpha} \in \mathbb{R}^L$ assigns an importance weight to each token. The pooled representation is then obtained by

$$\boldsymbol{o} = \sum_{i=1}^L \alpha_i (X)_{i,:}. \tag{10}$$

Given the inherently local nature of MEs, only partial facial action tokens truly activate the facial muscle movements of an ME, while the remaining tokens may have little or no effect on facial muscle movements. Therefore, the model should focus on these informative tokens rather than distributing attention uniformly across all tokens. To encourage sparsity and focus on informative local cues, we regularize $\boldsymbol{\alpha}$ with an entropy penalty:

$$\mathcal{L}_{\text{sparse}} = -\frac{1}{L} \sum_{i=1}^L \alpha_i \log \alpha_i. \tag{11}$$

Finally, the pooled feature $\boldsymbol{o}$ is fed into a linear classifier for recognition.

### 3.4 EMOTION-DESCRIPTION-GUIDED CLIP ALIGNMENT

To explicitly bridge the learned facial action tokens with human-understandable emotions, we incorporate an emotion-description-guided CLIP (EDCLIP) alignment. First, we design a textual action description for each expression class, as illustrated in Table 1. These descriptions are encoded by a frozen pretrained CLIP text encoder (Radford et al., 2021) to obtain class-level embeddings $\{\boldsymbol{t}_c\}_{c=1}^C$. Given an input ME, the pooled action feature $\boldsymbol{o}$ is mapped into the same feature space via a learnable function $f(\cdot)$. We then compute a contrastive loss between the image and text embeddings to align the action tokens with semantic emotion descriptions:

$$\mathcal{L}_{\text{CLIP}} = -\log \frac{\exp\left(\text{sim}(f(\boldsymbol{o}), \boldsymbol{t}_y)/\tau\right)}{\sum_{c=1}^C \exp\left(\text{sim}(f(\boldsymbol{o}), \boldsymbol{t}_c)/\tau\right)}, \tag{12}$$

where $\boldsymbol{t}_y$ is the text embedding of the ground-truth class, $\text{sim}(\cdot, \cdot)$ denotes cosine similarity, and $\tau$ is a temperature parameter.

Table 1: Text descriptions provided for each emotion category involved in our experimental setup.

| Emotion | Description |
|---------|-------------|
| Happiness (Positive) | A happy (positive) expression with raised cheeks and pulled lip corners. |
| Surprise | A surprised expression with raised eyebrows and an open mouth. |
| Disgust | A disgusted expression with lowered eyebrows, tightened eyelids, a wrinkled nose, and a raised upper lip. |
| Repression | A repressed expression with depressed lip corners, and a slight chin raise. |
| Negative | A negative expression with either lowered eyebrows, tightened eyelids, a raised upper lip, depressed lip corners, or a slight chin raise. |

However, MER datasets often contain a class, "others," which contains ambiguous MEs that are difficult to assign a clear emotion label. For this special case, no textual prompt is provided. Instead, we encourage the representation to be dissimilar from all textual embeddings of the specific emotion categories via a margin-based constraint:

$$\mathcal{L}_{\text{oth}} = \max\left(0, \max_{c \in \mathcal{C} \setminus \{c_{\text{oth}}\}} \text{sim}(f(\boldsymbol{o}), \boldsymbol{t}_c) - \delta\right), \tag{13}$$

where $\delta$ is a margin hyper-parameter, $\mathcal{C}$ is a set of all classes, and $c_{\text{oth}}$ is the "others" class. This constraint ensures that the "others" representations are not aligned excessively close to any specific emotion description, thus preserving their inherent ambiguity.

Therefore, the overall loss for EDCLIP is

$$\mathcal{L}_{\text{EDCLIP}} = \mathcal{L}_{\text{CLIP}} + \lambda_{\text{oth}}\mathcal{L}_{\text{oth}}, \tag{14}$$

where $\lambda_{\text{oth}}$ balances the contribution of $\mathcal{L}_{\text{oth}}$. Finally, the overall training objective combines the classification loss, the reconstruction regularizer, the sparsity penalty, and the EDCLIP loss:

$$\mathcal{L} = \mathcal{L}_{\text{cls}} + \lambda_{\text{rec}}\mathcal{L}_{\text{rec}} + \lambda_{\text{sparse}}\mathcal{L}_{\text{sparse}} + \lambda_{\text{EDCLIP}}\mathcal{L}_{\text{EDCLIP}}, \tag{15}$$

where $\mathcal{L}_{\text{cls}}$ denotes the cross-entropy loss for MER. $\lambda_{\text{rec}}$, $\lambda_{\text{sparse}}$, and $\lambda_{\text{EDCLIP}}$ are empirically set hyper-parameters.

### 3.5 Pretraining Details

We pretrain the C-VQ-VAE on the large-scale facial video dataset VoxCeleb (Nagrani et al., 2017). The VoxCeleb dataset consists of unconstrained interview videos collected from YouTube, where the frame rates vary depending on the source video (most videos are around 25–30 fps). For pretraining, we randomly sample image pairs with a temporal gap between 8 and 15 frames, which approximately corresponds to the typical duration of MEs (0.25–0.5 seconds). This sampling strategy yields about one million facial image pairs from more than 7,000 distinct identities, producing a rich diversity of motion patterns that greatly enhances the robustness of the learned facial action tokens (see Figure 9 in Appendix A.4 for sample pairs). During pretraining, we train the model for 800,000 steps with a batch size of 64. The hidden dimension of the network and the dimensionality of feature vectors in the codebook are set to 512 and 32, respectively.

## 4 Experiments

### 4.1 Datasets and Evaluation Metrics

We evaluate performance under the leave-one-subject-out (LOSO) protocol using Accuracy, UAR, and UF1 on four datasets: CASME-II (Yan et al., 2014), SMIC-HS (Li et al., 2013), SAMM (Davison et al., 2016), and CAS(ME)³ (Li et al., 2023). CASME-II contains 249 clips from 26 subjects recorded at 200 fps. SMIC-HS includes 164 clips from 16 subjects at 100 fps and $640 \times 480$ resolution; due to the lack of apex annotations, the middle frame is used as a pseudo-apex. SAMM comprises 159 clips from 32 subjects at 200 fps and $2040 \times 1088$ resolution. CAS(ME)³ provides 921 clips at 30 fps and $1280 \times 720$ resolution, making it one of the largest MER benchmarks.

Table 2: Comparison with the state-of-the-art methods on the composite dataset under the CDE protocol (See et al., 2019). The best results are highlighted in **bold** and the second-best results are marked with an underline.

| Methods | Full | | CASME-II | | SMIC-HS | | SAMM | |
|---|---|---|---|---|---|---|---|---|
| | UF1 | UAR | UF1 | UAR | UF1 | UAR | UF1 | UAR |
| OFF-ApexNet (Gan et al., 2019) | 0.7196 | 0.7096 | 0.8764 | 0.8681 | 0.6817 | 0.6695 | 0.5409 | 0.5392 |
| STSTNet (Liong et al., 2019) | 0.7353 | 0.7605 | 0.8382 | 0.8686 | 0.6801 | 0.7013 | 0.6588 | 0.6810 |
| Graph-AU (Lei et al., 2021) | 0.7914 | 0.7933 | 0.8798 | 0.8710 | 0.7192 | 0.7215 | 0.7751 | 0.7890 |
| SLSTT (Zhang et al., 2022) | 0.8160 | 0.7900 | 0.9010 | 0.8850 | 0.7400 | 0.7200 | 0.7150 | 0.6430 |
| FRL-DGT (Zhai et al., 2023) | 0.8120 | 0.8110 | 0.9190 | 0.9030 | 0.7430 | 0.7490 | 0.7720 | 0.7580 |
| SRMCL (Bao et al., 2024) | 0.8630 | 0.8830 | 0.9635 | 0.9649 | 0.7946 | 0.8053 | 0.8470 | **0.8866** |
| MFDAN (Cai et al., 2024) | 0.8453 | 0.8688 | 0.9134 | 0.9326 | 0.6815 | 0.7043 | 0.7871 | 0.8196 |
| HTNet (Wang et al., 2024) | 0.8603 | 0.8475 | 0.9532 | 0.9516 | 0.8049 | 0.7905 | 0.8131 | 0.8124 |
| LTR3O (Zhu et al., 2025) | 0.8931 | 0.8819 | 0.9578 | 0.9487 | 0.8336 | 0.8298 | **0.8912** | 0.8526 |
| **Ours** | **0.8943** | **0.8967** | **0.9738** | **0.9754** | **0.8422** | **0.8476** | 0.8716 | 0.8513 |

## 4.2 TRAINING DETAILS

The Transformer for MER consists of 2 standard Transformer encoder layers with a hidden dimension of 512 and 8 attention heads. The model is trained with the Adam optimizer (Kingma & Ba, 2015) on each dataset for 120 epochs, using a learning rate of $2.0 \times 10^{-4}$ and a weight decay of $1.0 \times 10^{-5}$, together with C-VQ-VAE finetuning. The coefficient $\delta$ in Eq. (13) is set to 0.2. The loss weights $\lambda_{\text{oth}}, \lambda_{\text{rec}}, \lambda_{\text{sparse}}, \lambda_{\text{EDCLIP}}$ are set to 0.2, 0.1, $1.0 \times 10^{-3}$, and 0.3, respectively.

## 4.3 COMPARISON WITH STATE-OF-THE-ART METHODS

We first compare our method with state-of-the-art (SOTA) methods. Table 2 reports the results under the Composite Database Evaluation (CDE) protocol proposed in the MEGC 2019 Challenge (See et al., 2019), where CASME-II, SMIC-HS, and SAMM are combined to enrich samples for three-class recognition. Our method achieves the highest UF1 score of 0.8943 and UAR of 0.8967 for the overall performance, outperforming other pixel-level motion-based methods. These results highlight the strong capability of the proposed facial action representations extracted from D-FACE. Note that our method achieves the second-best UF1 and third-best UAR on SAMM, due to the limitation that images in SAMM are grayscale, whereas our model was pretrained on RGB images, which may affect facial action extraction.

We further evaluate our method on multi-class scenarios. Table 3 show the results on CASME-II for five-class recognition. Despite the limited number of annotated samples, our method achieves competitive performance, ranking second overall and surpassing other methods based on estimated optical flow. We also report results on the CAS(ME)[3] dataset, which contains 921 ME clips across four classes. As shown in Table 4, our method outperforms the second-best method, MER-CLIP (Liu et al., 2025), which

Table 3: Comparison with the state-of-the-art methods on CASME-II.

| Method | CASME-II (5-class) | | |
|---|---|---|---|
| | ACC | UF1 | UAR |
| Graph-TCN (Lei et al., 2020) | 0.7246 | 0.7398 | - |
| MMNet (Li et al., 2022) | **0.8835** | **0.8676** | - |
| FRL-DGT (Zhai et al., 2023) | 0.7570 | 0.7480 | 0.7350 |
| $\mu$-bert (Nguyen et al., 2023) | - | 0.8553 | 0.8348 |
| SRMCL (Bao et al., 2024) | 0.8320 | 0.8286 | - |
| LTR3O (Zhu et al., 2025) | 0.8178 | 0.7905 | - |
| SODA (Zhang et al., 2025) | 0.8418 | 0.8141 | - |
| MER-CLIP (Liu et al., 2025) | 0.8233 | 0.8378 | - |
| **Ours** | 0.8474 | 0.8571 | **0.8694** |

Table 4: Comparison with the state-of-the-art methods on CAS(ME)[3].

| Method | CAS(ME)[3] (4-class) | |
|---|---|---|
| | UF1 | UAR |
| $\mu$-bert (Nguyen et al., 2023) | 0.4718 | 0.4913 |
| SFAMNet (Liong et al., 2024) | 0.4462 | 0.4797 |
| MER-CLIP (Liu et al., 2025) | 0.6544 | 0.6242 |
| **Ours** | **0.6807** | **0.6469** |

additionally uses action unit annotations, by 4.02% in UF1 and 3.64% in UAR, highlighting our robustness in complex multi-class scenarios.

## 4.4 C-VQ-VAE CAPACITY STUDIES

We conduct capacity studies on the codebook size and sequence length of the C-VQ-VAE on the CAS(ME)$^3$ dataset, as summarized in Table 5. The codebook size specifies the number of feature vectors in the codebook, whereas the sequence length denotes the number of facial action tokens that represent the muscle movements between two facial images. The results indicate that the optimal MER performance is achieved with a codebook size of 32 and a sequence length of 16. Smaller values fail to capture sufficient details of ME motion, whereas larger values lead to redundancy. An overly large codebook destabilizes training and harms MER performance, where many feature vectors in the codebook become infrequently used or capture unnecessary details rather than essential facial motions, resulting in less meaningful representations. This suggests that a compact yet expressive tokenization is crucial for capturing subtle ME motion without overwhelming the subsequent Transformer with noisy or redundant information.

Table 5: Capacity studies on the codebook size and sequence length of the C-VQ-VAE.

| BookSize | SeqLen | CAS(ME)$^3$ (4-class) | |
|---|---|---|---|
| | | UF1 | UAR |
| 16 | 9 | 0.5210 | 0.5217 |
| 16 | 16 | 0.6726 | 0.6378 |
| **32** | **16** | **0.6807** | **0.6469** |
| 64 | 16 | 0.6227 | 0.5963 |
| 128 | 16 | 0.6399 | 0.6017 |
| 16 | 25 | 0.6481 | 0.6177 |
| 32 | 25 | 0.6578 | 0.6320 |
| 64 | 25 | 0.6423 | 0.6071 |

Table 6: Ablation studies on the choice of recognition network, feature aggregation strategy, and EDCLIP loss on CASME-II for five-class classification.

| Category | Method | Acc | UF1 | UAR |
|---|---|---|---|---|
| Recognition Network | CNN Backbone | 0.8313 | 0.8120 | 0.8204 |
| | Transformer with 2D PE | 0.8327 | 0.8164 | 0.8234 |
| | Transformer with 1D PE (ours) | **0.8474** | **0.8571** | **0.8694** |
| Feature Aggregation | Average Pooling | 0.8434 | 0.8345 | 0.8374 |
| | Class Token | 0.8313 | 0.8294 | 0.8593 |
| | Sparse Attention Pooling (ours) | **0.8474** | **0.8571** | **0.8694** |
| EDCLIP Loss | w/o EDCLIP | 0.8394 | 0.8286 | 0.8384 |
| | w/ EDCLIP (ours) | **0.8474** | **0.8571** | **0.8694** |

## 4.5 ABLATION STUDIES

We conduct ablation studies to validate the effectiveness of each component in our framework. The results are shown in Table 6.

**Recognition Network.** To verify the correctness of our empirical observations on facial action tokens, we replace the Transformer with 1D positional embeddings (PE) by either a CNN backbone or a Transformer with 2D PE, while keeping the same number of network layers. The results show that the Transformer with 1D PE outperforms both alternatives, demonstrating the advantage of 1D sequential modeling over 2D spatial structures in capturing order-dependent token semantics.

**Sparse Attention Pooling.** Next, we examine the contribution of sparse attention pooling by comparing it with two alternatives: average pooling and the class-token mechanism. The results show that sparse attention pooling yields better performance, confirming that only a subset of action tokens is informative and should be emphasized for MER.

**EDCLIP Loss.** Finally, we evaluate the effectiveness of the EDCLIP loss. We compare models trained with and without the EDCLIP loss. Results show that incorporating emotion descriptions

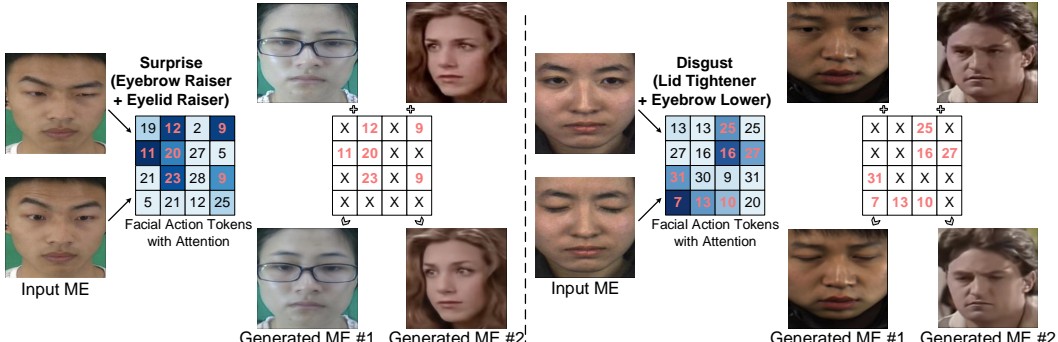

Figure 4: Visualization of sparse attention pooling and facial action tokens. For each sample, the action information between onset and apex frames are encoded into action tokens, weighted by sparse attention pooling (darker indicates higher weight). The salient tokens are then applied to cross-identity and cross-domain faces, generating new facial images that preserve the same underlying MEs.

further improves both accuracy and balanced metrics, showing that bridging action tokens to emotion semantics provides additional supervision and enhances the discriminability of learned features.

## 4.6 QUALITATIVE ANALYSES

In addition to quantitative results, we provide qualitative analysis to demonstrate the effectiveness of sparse attention pooling and the generalization of our facial action tokens. The visualizations are presented in Figure 4. For a given ME, we first visualize the index map of facial action tokens with attention scores, which highlight the critical tokens for recognition. To further examine their validity and generalization ability, we perform controlled manipulation experiments by applying the salient action tokens to 1) an image from a different subject; and 2) an in-the-wild face collected from unconstrained conditions. For each sample, we first use two identical facial images to determine the index map corresponding to the "no action" state. Then, we selectively replace only the indices emphasized by sparse attention pooling with those from a real ME pair, while keeping the rest unchanged. The reconstructed results consistently preserve the underlying identity while presenting the intended expression, validating both the effectiveness of sparse attention pooling and the cross-subject, cross-domain generalization ability of our learned facial action tokens.

In addition, we analyze the facial action tokens in the learned codebook and observe several interesting findings. Specifically, tokens with higher similarity in the feature space tend to correspond to similar motion semantics, while variations in feature distance are related to differences in expression intensity. Based on this insight, we further investigate: 1) diverse ME generation with various intensities, and 2) the interpretability of the codebook for controllable expression generation. More detailed analyses and visualizations are provided in Appendices A.1 and A.2.

## 5 CONCLUSION

In this paper, we introduced D-FACE, a Discrete Facial ACtion Encoding framework pretrained on large-scale facial video data, to discretize facial muscle movements between two facial images into identity- and domain-invariant facial action tokens. Through empirical analyses, we observed that these action tokens exhibit order-dependent semantics and can be organized as one-dimensional sequences, which motivated the use of a Transformer with sparse attention pooling to capture informative local cues for MER. Furthermore, we incorporated an emotion-description-guided CLIP (EDCLIP) alignment to explicitly bridge action tokens with human-understandable emotions. In EDCLIP, the "others" category, which lacks appropriate textual prompts due to its ambiguity, is pushed away from other textual embeddings. Extensive experiments on multiple MER datasets demonstrated that D-FACE not only achieves state-of-the-art performance but also exhibits strong cross-subject and cross-domain generalization, including the ability for ME generation. These results suggested that shifting from pixel-level motion descriptors to robust semantic-level action tokens offers a promising new perspective for ME analysis.

ACKNOWLEDGMENTS

This work was supported by Innovation Platform for Society 5.0 from Japan Ministry of Education, Culture, Sports, Science and Technology, and JSPS KAKENHI Grant Number JP24K03010.

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

# A APPENDIX

## A.1 DIVERSE MICRO-EXPRESSION GENERATION

We further investigate the generation of diverse MEs with various intensities. To this end, we analyze the feature similarity and distance among the vectors in the codebook, as illustrated in Figures 5 and 6. The results indicate that certain feature vectors exhibit high similarity and low distance, suggesting they convey highly similar semantics. We hypothesize that tokens with high similarity share the same facial motion semantics, while feature distance regulates expression intensity. Quantitative experiments are conducted to verify this hypothesis, with results shown in Figure 7. These results demonstrate that diverse ME generation with various intensities can be achieved by replacing a specific facial action token with another highly similar token.

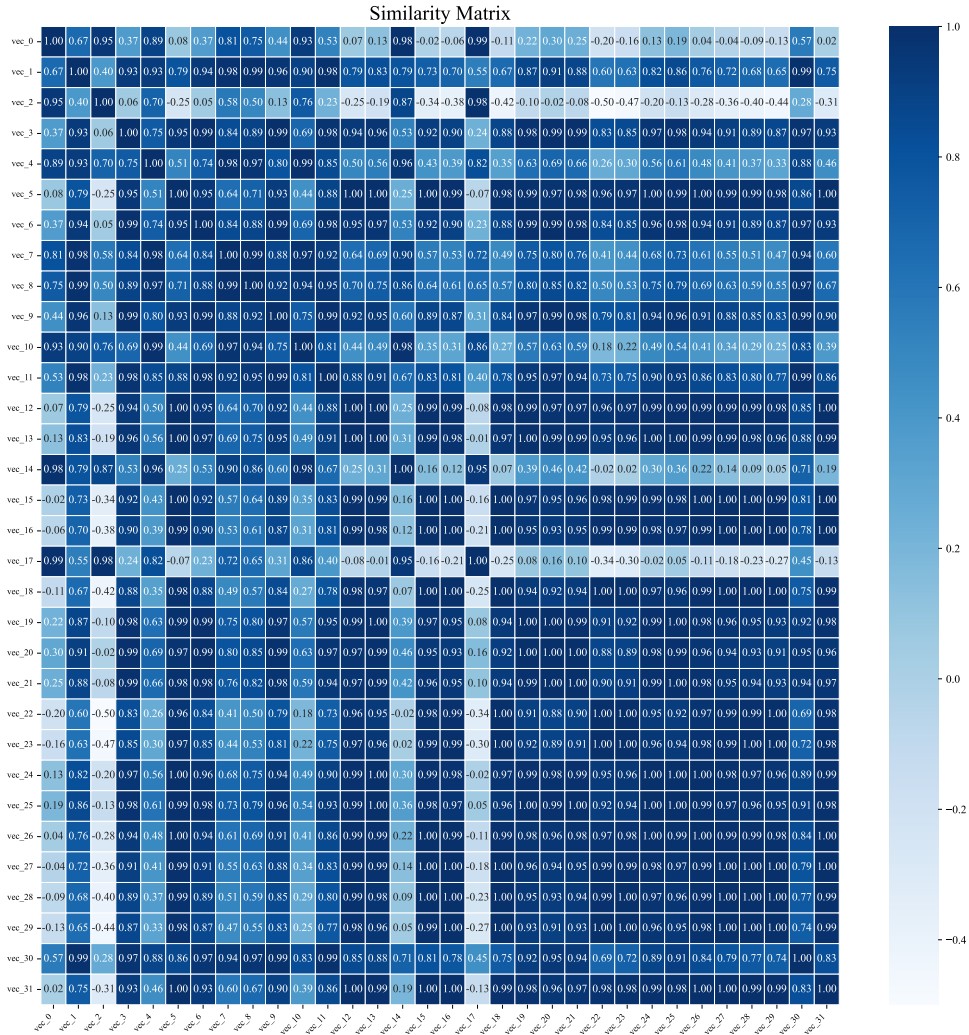

Figure 5: Cosine similarity among the feature vectors in the codebook.

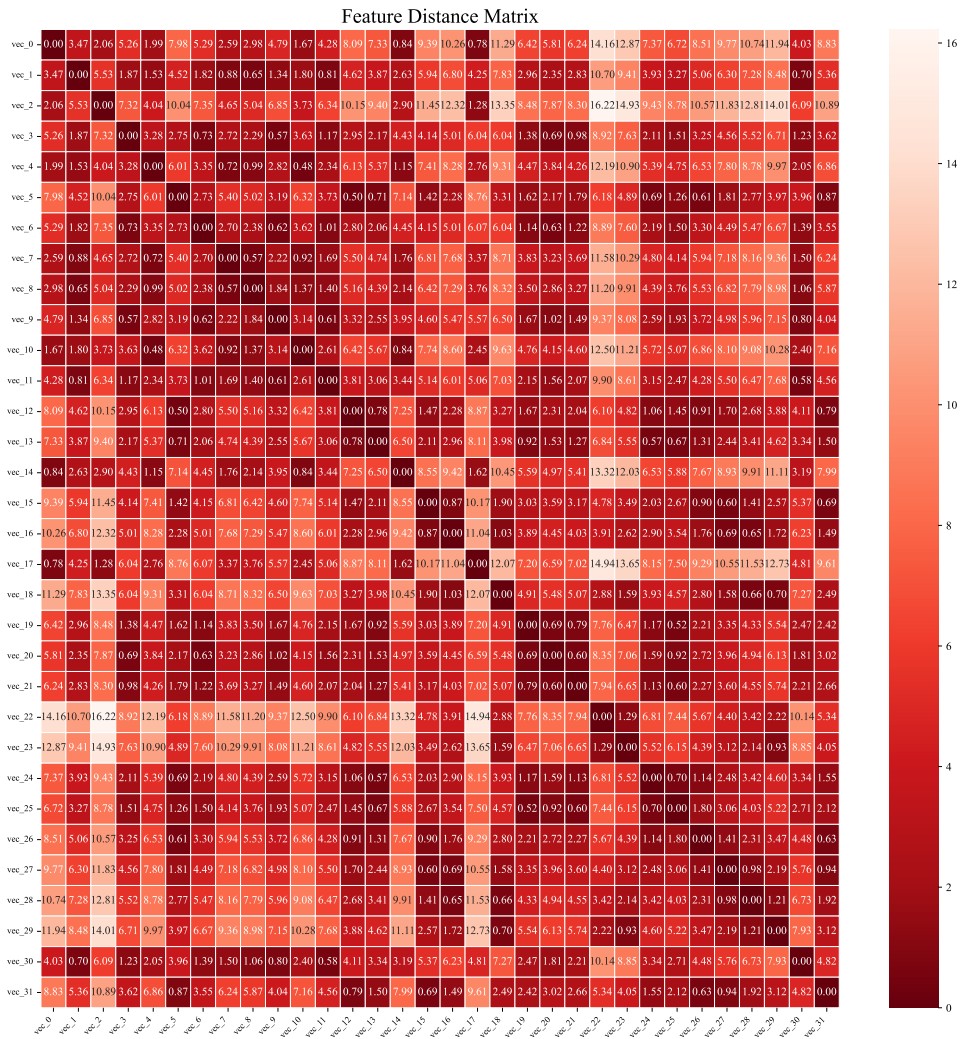

Figure 6: Euclidean distance among the feature vectors in the codebook.

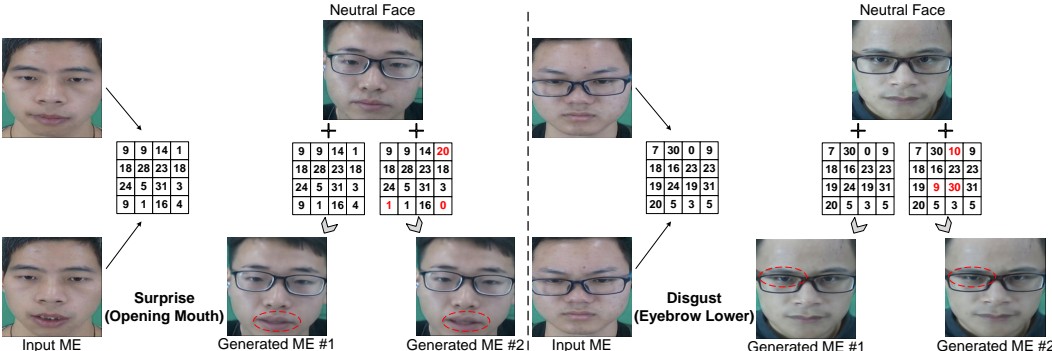

Figure 7: Visualization of intensity control in ME generation. For each sample, we first extract facial action tokens from an ME clip. These tokens are then applied to a neutral face to generate a facial image that presents the same ME. By selectively replacing certain action tokens with highly similar alternatives, we generate a new facial image that preserves the original ME while exhibiting different intensity.

A.2  INTERPRETABILITY OF THE FACIAL ACTION TOKENS

In this subsection, we further investigate the interpretability of the learned facial action tokens. Using the similarity matrix (Figure 5) and the feature distance matrix (Figure 6), we cluster the 32 codebook vectors into 6 groups with consistent semantics, as summarized in Table 7. To examine the motion semantics of each token, we perform a systematic $6 \times 16$ manipulation study, where each cluster is substituted at every sequence location of the index map. Specifically, we first generate a "no action" index map from two identical facial images. Then, by altering one cluster at a time while keeping all others fixed, we observe the resulting facial image. The summarized observations are presented in Table 8.

Table 7: Clustering results of the codebook feature vectors.

| Cluster | Indices |
|---------|---------|
| 0 | 1, 3, 6, 9, 11, 30 |
| 1 | 5, 12, 15, 16, 26, 27, 31 |
| 2 | 0, 2, 14, 17 |
| 3 | 18, 22, 23, 28, 29 |
| 4 | 13, 19, 20, 21, 24, 25 |
| 5 | 4, 7, 8, 10 |

Table 8: The emotion-related motion semantics of each token at each sequence location of the index map. The motions are described approximately in terms of Action Units (AUs) (Ekman & Rosenberg, 1997). "Other AUs" indicates AUs not directly involved in MEs (e.g., eyeball movements, head movements).

| Clus. Pos. | 0 | 1 | 2 | 3 | 4 | 5 |
|---|---|---|---|---|---|---|
| (0, 0) | Other AUs | Other AUs | Other AUs | Other AUs | Other AUs | No action |
| (0, 1) | No action | Other AUs | Other AUs | Other AUs | Other AUs | Other AUs |
| (0, 2) | Slight 4+7 | Marked 4+7 | No action | Strong 4+7 | Slight 4+7 | Trace 4+7 |
| (0, 3) | No action | Other AUs | Other AUs | Other AUs | Other AUs | Other AUs |
| (1, 0) | Slight 1+2+12 | Trace 1+2+12 | Strong 1+2+12 | No action | Marked 1+2+12 | Slight 1+2+12 |
| (1, 1) | Other AUs | Other AUs | Other AUs | No action | Other AUs | Other AUs |
| (1, 2) | Other AUs | Other AUs | Other AUs | No action | Other AUs | Other AUs |
| (1, 3) | Slight 12 | Trace 12 | Strong 12 | No action | Marked 12 | Slight 12 |
| (2, 0) | Trace 5+12 | Trace 7+23 | Marked 5+12 | Marked 7+23 | No action | Slight 5+12 |
| (2, 1) | Slight 4+7 | No action | Strong 4+7 | Trace 1+2+5 | Trace 4+7 | Marked 4+7 |
| (2, 2) | Slight 10 | No action | Strong 10 | Trace 24 | Trace 10 | Marked 10 |
| (2, 3) | Slight 24 | No action | Strong 24 | Trace 10 | Trace 24 | Marked 24 |
| (3, 0) | Slight 7 | Slight 5 | Strong 7 | Strong 5 | No action | Marked 7 |
| (3, 1) | Slight 25 | Slight 24 | Strong 25 | Strong 24 | No action | Marked 25 |
| (3, 2) | Slight 7+24 | Slight 5+25 | Strong 7+24 | Strong 5+25 | No action | Marked 7+24 |
| (3, 3) | Slight 7+26 | Slight 5+17 | Strong 7+26 | Strong 5+17 | No action | Marked 7+26 |

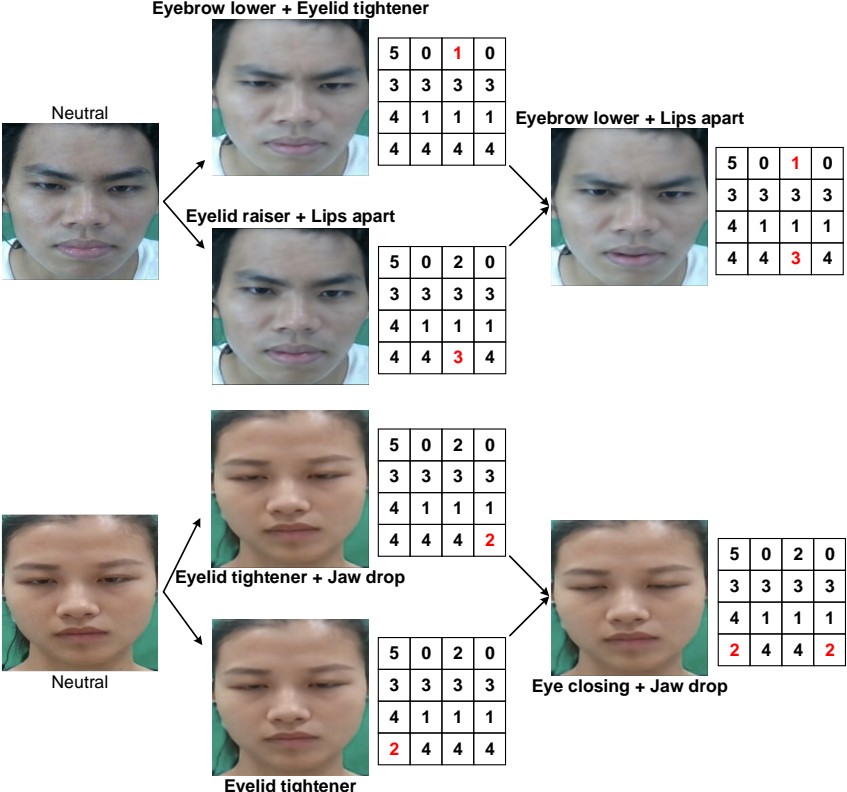

Figure 8: Visualization of controllable expression generation. For each sample, we show the neutral face (left), two generated facial expressions with a single token replaced (middle), and the composite facial expression (right).

We conduct qualitative experiments to validate our observations through controllable expression generation. Some representative results are shown in Figure 8, indicating that different facial action tokens can be flexibly combined to generate diverse facial expressions. These results further support that token interactions rely on order-dependent and global contextual modeling rather than 2D position-dependent structure.

### A.3 SENSITIVITY ANALYSIS OF TRANSFORMER HYPERPARAMETERS

To evaluate the stability of the Transformer for MER, we conducted a sensitivity analysis on two key architectural hyperparameters: the number of attention layers $L_{attn}$ and the hidden dimensionality $d$. Specifically, we evaluated models with {1, 2, 3, 4} layers and hidden dimensionality {256, 512, 768, 1024} while keeping all other settings unchanged.

As shown in Table 9, the overall performance on the composite dataset under the CDE protocol dataset show that the overall performance varies only slightly across different settings, and all configurations achieve comparable accuracy. The model with 2 layers and a hidden size of 512 offers a good trade-off between efficiency and performance, which we therefore adopt in the main experiments. These observations indicate that our

Table 9: Ablation study of Transformer depth $L_{attn}$ and hidden size $d$ under the CDE protocol.

| $L_{attn}$ | $d$ | Composite | |
|---|---|---|---|
| | | UF1 | UAR |
| 1 | 512 | 0.8789 | 0.8927 |
| 2 | 512 | **0.8943** | 0.8967 |
| 3 | 512 | 0.8905 | **0.9032** |
| 4 | 512 | 0.8816 | 0.8967 |
| 2 | 256 | 0.8764 | 0.8877 |
| 2 | 512 | **0.8943** | 0.8967 |
| 2 | 768 | 0.8798 | **0.8990** |
| 2 | 1024 | 0.8717 | 0.8734 |

results are insensitive to reasonable hyperparameter changes. These results also demonstrate that our framework is a stable and reproducible pipeline.

### A.4 EXAMPLE SAMPLES FROM THE PRETRAINING DATASET

In this subsection, we visualize several examples of frame pairs obtained through our sampling strategy from the large-scale facial video dataset VoxCeleb, as shown in Figure 9. These examples provide an intuitive illustration of the diversity present in our pretraining data, including variations in identity, expression intensity, and recording conditions. Such diversity is essential for enabling the C-VQ-VAE codebook to learn robust and generalizable semantic motion representations.

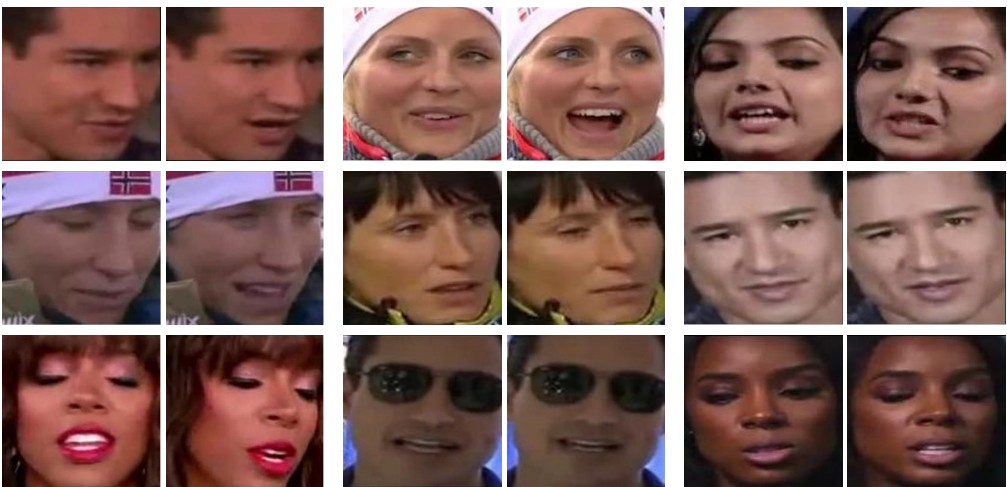

Figure 9: Example frame pairs sampled from the VoxCeleb pretraining dataset.

### A.5 ABLATION STUDY ON THE CHOICE OF TEXT PROMPTS IN EDCLIP.

To validate the design of our EDCLIP, we conduct an ablation study on the choices of text prompts used for aligning action tokens with emotion semantics. We compare our default prompt, "A [emotion] expression with [general facial action descriptions].", against two alternative templates: 1) "A [emotion] face showing [general facial action descriptions]."; and 2) "A [emotion] expression with [AU annotations of each sample].", where the general action descriptions are replaced with AU-based descriptions following MER-CLIP (Liu et al., 2025). For the AU-based setting, samples belonging to the same emotion category may correspond to different AU combinations and thus different textual prompts.

Table 10: Ablation study on the text prompts used in EDCLIP. Variant #1 corresponds to "A [emotion] face showing [general facial action descriptions].", while Variant #2 corresponds to "A [emotion] expression with [AU annotations of each sample].", where the AU descriptions follow MER-CLIP (Liu et al., 2025).

| Template | CASME-II (5-class) | | |
| --- | --- | --- | --- |
| | ACC | UF1 | UAR |
| Variant #1 | **0.8514** | 0.8565 | 0.8617 |
| Variant #2 | 0.8273 | 0.8255 | 0.8231 |
| Ours | 0.8474 | **0.8571** | **0.8694** |

The results are shown in Table 10. We observe that the general-action–based variant yields performance close to our default template, indicating that EDCLIP is not overly sensitive to minor template changes and remains stable and effective. In contrast, the AU-based prompts lead to performance degradation. The reason is that AU-based descriptions vary across samples even within the same emotion class. Consequently, each emotion category no longer shares a consistent textual embedding but instead corresponds to many distinct AU-based textual vectors. This removes the class-level semantic anchor in the text embedding space and forces samples of the same emotion to be aligned toward different textual directions. Such misalignment weakens the contrastive learning objective, destabilizes training, and ultimately harms recognition performance in our framework.

In contrast, our original design assigns one textual prompt to each emotion class by combining the emotion name with general facial action descriptions. This ensures that all samples of the same emotion are pulled toward a single, stable semantic embedding, providing a strong and consistent anchor for alignment. The ablation results further support that our facial action tokenizer already learns robust and discriminative motion representations without requiring precise AU annotations.

### A.6 QUALITATIVE EVALUATION ON IN-THE-WILD AND LOW-RESOLUTION FACES

To evaluate the cross-domain generalization ability of D-FACE, we conduct qualitative experiments on a collection of in-the-wild facial images obtained from unconstrained online sources. For each facial image, we first generate the token map that corresponds to the "no-action" state, and then selectively alter token indices to generate new facial images.

To further evaluate robustness under low-quality inputs, we downsample each facial image to 20% of its original resolution and then feed the degraded images into the facial action tokenizer. We repeat the same procedure: obtaining the "no-action" token map and applying the same token manipulations for facial image generation.

As shown in Figure 10, D-FACE generalizes well on both high-resolution and low-quality in-the-wild images, generating natural faces with diverse facial actions. These results demonstrate that the learned facial action tokens encode stable and meaningful motion semantics that transfer reliably across identities, domains, and even under substantial resolution degradation.

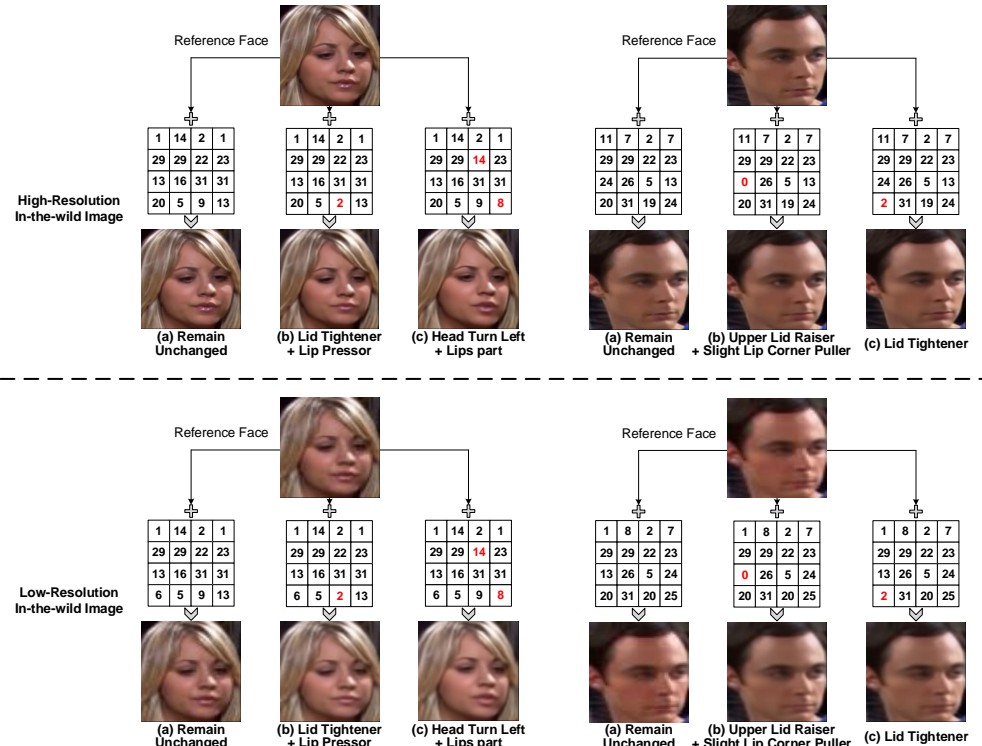

Figure 10: Qualitative evaluation on high- and low-quality in-the-wild facial images. The numbers in red stand for the altered indices.

### A.7 EVALUATION ON MICRO-EXPRESSION ACTION UNIT RECOGNITION

To further evaluate the robustness of the learned facial action tokens, we extend our framework beyond MER and conduct ME AU recognition, which provides a direct evaluation of the quality of the motion representations. Following the evaluation protocol of Li et al. (2021), we perform AU recognition on the CASME-II dataset and report the F1-scores for AU1, AU2, AU4, AU7, AU12, AU14, AU15, and AU17, as well as the average F1-score. In this experiment, we use the same pretrained facial

Table 11: Evaluation of micro-expression action unit recognition on the CASME-II dataset.

| Method | AU1 | AU2 | AU4 | AU7 | AU12 | AU14 | AU15 | AU17 | AVG |
|---|---|---|---|---|---|---|---|---|---|
| Li et al. (2021) | 0.726 | 0.721 | **0.898** | 0.569 | **0.796** | **0.685** | 0.715 | 0.700 | 0.726 |
| Ours | **0.902** | **0.868** | 0.849 | **0.660** | 0.667 | 0.636 | **0.828** | **0.800** | **0.776** |

action tokenizer and the same Transformer architecture with identical hyperparameter settings, while removing the EDCLIP module since AU recognition does not require aligning motion representations with textual emotion semantics.

As shown in Table 11, our framework achieves an average F1-score of 0.776 on CASME-II, even without task-specific considerations and module designs. We compare our results with the most recent work published in established venues. Despite not being designed for AU recognition, the proposed semantic-level facial action tokens remain robust and effective, demonstrating its transferability across tasks.

## A.8 ABLATION STUDY ON LOSS WEIGHTING

We also conducted ablation studies on the loss weights of the EDCLIP loss (i.e., $\lambda_{\text{EDCLIP}}$) and the margin-based constraint for the "others" category (i.e., $\lambda_{\text{oth}}$) to determine the optimal hyperparameters. The results are shown in Table 12. For $\lambda_{\text{EDCLIP}}$, we find that a small weight leads to insufficient contrastive alignment, while an overly large weight affects the classification objective. For $\lambda_{\text{oth}}$, we observe that overly large values put too much emphasis on separating the "others" samples from all textual embeddings, thus leading to performance degradation. Based on these observations, we set $\lambda_{\text{EDCLIP}} = 0.3$ and $\lambda_{\text{oth}} = 0.2$ for all experiments.

Table 12: Ablation study on loss weighting.

| $\lambda_{\text{EDCLIP}}$ | CASME-II (5-class) | | |
|---|---|---|---|
| | Acc | UF1 | UAR |
| 0.0 | 0.8394 | 0.8286 | 0.8384 |
| 0.1 | 0.8434 | 0.8409 | 0.8604 |
| 0.3 | **0.8474** | **0.8571** | **0.8694** |
| 0.5 | **0.8474** | 0.8455 | 0.8631 |
| 0.7 | 0.8394 | 0.8353 | 0.8429 |

| $\lambda_{\text{oth}}$ | Acc | UF1 | UAR |
|---|---|---|---|
| 0.0 | 0.8474 | 0.8472 | 0.8643 |
| 0.2 | 0.8474 | **0.8571** | **0.8694** |
| 0.4 | **0.8514** | 0.8356 | 0.8533 |
| 0.6 | 0.8434 | 0.8365 | 0.8438 |

## A.9 EFFECT OF THE DISCRETIZATION BOTTLENECK IN VQ-VAE

To examine whether the discretization bottleneck in VQ-VAE loses fine-grained motion nuances crucial for MER, we conduct an additional experiment using the same pretrained C-VQ-VAE. As in our main framework, we first compute the feature difference between the two input frames, but we remove the quantization step and keep the motion features in a continuous latent space.

Table 13: Evaluation of the discretization process in VQ-VAE on the composite dataset under the CDE protocol.

| Method | Composite | |
|---|---|---|
| | UF1 | UAR |
| w/o discretization | 0.8762 | 0.8799 |
| w/ discretization (Ours) | **0.8943** | **0.8967** |

As shown in Table 13, removing discretization leads to a clear drop in MER performance, even though discretization may theoretically introduce some loss of reconstruction details. This result confirms that the quantized representations are cleaner and more motion-focused, effectively filtering out irrelevant appearance variations, which is beneficial for MER.

## A.10 THE USE OF LARGE LANGUAGE MODELS (LLMS)

We used LLMs for grammar checking.

