# OpenReview forum: "From Pixels to Semantics: Unified Facial Action Representation Learning for Micro-Expression Analysis"
_ICLR.cc/2026/Conference — ICLR 2026 Poster_

### Official Review · Reviewer_UyN7 · 2025-10-25

**Soundness:** 3
**Presentation:** 2
**Contribution:** 2
**Rating:** 6
**Confidence:** 4

**Summary:**

The paper introduces D-FACE, a Discrete Facial ACtion Encoding framework that shifts micro-expression recognition (MER) from pixel-level motion descriptors (e.g., optical flow) to semantic-level facial action tokens. Using a conditional vector-quantized VAE, D-FACE discretizes facial muscle movements into domain- and identity-invariant “action tokens.” These tokens are treated as one-dimensional sequences and processed with a Transformer featuring sparse attention pooling to capture localized, discriminative cues. An additional emotion-description-guided CLIP (EDCLIP) alignment connects visual token features with textual emotion descriptions, enhancing semantic interpretability. Experiments on several benchmark datasets show that D-FACE achieves competitive or superior accuracy and exhibits cross-subject and cross-domain generalization.

**Strengths:**

1. Proposes a novel paradigm shift for MER: from pixel-level motion descriptors to semantic-level, discrete facial action tokens, potentially improving robustness and transferability.

2. Designs a C-VQ-VAE–based facial action tokenizer, the first to discretize motion representations for micro-expressions. Introduces a Transformer with sparse attention pooling to focus on informative local cues, demonstrating improved interpretability and performance.

3. Develops EDCLIP alignment to connect visual action tokens with human-understandable emotion semantics, enhancing explainability.

4. Shows cross-domain and cross-identity generalization, as well as potential for facial expression generation, marking a broader contribution beyond recognition.

**Weaknesses:**

This paper has a few issues:
(1) Figure 1(b) (and related illustrations) are difficult to follow, limiting accessibility of the proposed conceptual shift, especially the interpretation of token map. A few more descriptive words would help.

(2) The novelty, while conceptually interesting, relies heavily on adapting existing VQ-VAE and CLIP mechanisms. In the introduction, it is better to further clarify how the method contributes novel insights on top of existing mechanisms.

(3) Logic clarity in parts (especially empirical analysis of tokens in Sec. 3.2) could be improved; the link between position-dependent semantics and 1D modeling is asserted but not quantitatively validated. Method correctness appears sound but somewhat over-engineered; e.g., the choice of hyperparameters and training stability of the C-VQ-VAE + Transformer pipeline is not discussed in detail.

(4) The reported accuracy improvements are marginal (e.g., small gains over LTR-3O and SRMCL in Table 2), raising questions about practical significance. There is little ablation or visualization proving that the semantic tokens truly capture meaningful, interpretable muscle movements.

**Questions:**

In addition to the questions relevant to weaknesses, there are a few more:
(1) How generalizable is D-FACE to in-the-wild, low-quality videos, given its dependence on large-scale pretraining on high-quality data?

(2) Could the discretization bottleneck of VQ-VAE lose fine-grained motion nuances crucial for micro-expression detection?

---

> ### Author Response · Authors · 2025-11-20
> **Response to Reviewer UyN7 (part 1)**
>
> Thank you very much for your positive evaluation. Please find our responses to each of your concerns below.
>
> **Weakness 1: Figure 1(b) (and related illustrations) are difficult to follow, limiting accessibility of the proposed conceptual shift, especially the interpretation of token map. A few more descriptive words would help.**
>
> **A:** Thank you for the suggestion. We have revised the caption of Figure 1(b) to a clearer and more intuitive description: (b) D-FACE encodes facial motions into facial action tokens, which are semantic motion representations that generalize across different identities and domains.
>
> **Weakness 2: The novelty, while conceptually interesting, relies heavily on adapting existing VQ-VAE and CLIP mechanisms. In the introduction, it is better to further clarify how the method contributes novel insights on top of existing mechanisms.**
>
> **A:** Thank you for your valuable suggestion. We have clarified more about our contributions and novel insights when adapt VQ-VAE and CLIP mechanisms to the MER task even facial motion analysis domain.
>
> For the EDCLIP, we revised the introductory paragraph as “**Furthermore, we introduce an emotion-description-guided CLIP (EDCLIP) alignment to align the learned action tokens with human-understandable emotions. Specifically, our textual prompts are constructed by combining each emotion name with its typical facial action descriptions, providing a shared and consistent textual embedding for all samples of the same emotion and enabling a more explicit bridge between motion representations and emotional semantics. In addition, MER uniquely includes an ambiguous “others” category that cannot be precisely described through language. To address this, we adapt the CLIP mechanism by enforcing that “others” samples remain semantically distant from all emotion-specific textual embeddings rather than being aligned to an ill-defined description, allowing EDCLIP to better fit the characteristics of the MER task.**”
>
> For the overall framework, we emphasized “**D-FACE achieves a paradigm shift for MER from observing pixel-level movements to modeling semantic-level action tokens. This is enabled by, for the first time, adapting VQ-VAE to MER through domain-specific considerations, carefully designed large-scale facial video pretraining, and extensive token–motion analysis that reveals interpretable and order-dependent token semantics.**” We mentioned the pretraining is “carefully designed” because at first, we attempted to pretrain the tokenizer on CAS(ME)³, a large-scale micro-expression dataset containing 80 hours of video and over 8 million frames, including more than 1,000 manually annotated micro-expressions and 3,400 macro-expressions. However, the pretrained results were unsatisfactory. The codebook failed to learn robust action representations. We attribute this to the fact that most image pairs with an 8–15 frame gap exhibit extremely subtle motion, providing insufficient motion diversity for effective codebook learning. Therefore, we employed VoxCeleb for pretraining. Our sampling strategy, which randomly selects two frames with an 8–15 frame temporal gap (corresponding to the typical ME duration), naturally produces over one million face pairs spanning more than 7,000 identities and covering both subtle and obvious facial motions. This leads to a much richer and more diverse set of motion patterns, which substantially improves the robustness of the learned action tokens.
>
> We believe these additions further clarify our contributions and highlight the new insights that guided the design of our framework for MER from a new perspective. We have included these additions to Section 1 Line 86–98. Please kindly refer to it.

---

> ### Author Response · Authors · 2025-11-20
> **Response to Reviewer UyN7 (part 2)**
>
> **Weakness 3: Logic clarity in parts (especially empirical analysis of tokens in Sec. 3.2) could be improved; the link between position-dependent semantics and 1D modeling is asserted but not quantitatively validated. Method correctness appears sound but somewhat over-engineered; e.g., the choice of hyperparameters and training stability of the C-VQ-VAE + Transformer pipeline is not discussed in detail.**
>
> **A:** Thank you for your valuable suggestion. For the logic clarity, we have not only included **more theoretical analysis** in the revised Section 3.2, but also included **quantitative comparison with “Transformer with 2D positional embeddings”** in the Table 6.
>
> Results on CASME-II (5-class)
> | Category             | Method                        | Acc     | UF1     | UAR     |
> |----------------------|-------------------------------|---------|---------|---------|
> | Recognition Network  | CNN Backbone                  | 0.8313  | 0.8120  | 0.8204  |
> |                      | Transformer with 2D PE        | 0.8327  | 0.8164  | 0.8234  |
> |                      | Transformer with 1D PE (ours) | **0.8474** | **0.8571** | **0.8694** |
>
>
> More analysis: “First, the same action token may activate different facial muscle motions when placed at different sequence locations, indicating that a token’s semantics depend on its sequence location rather than being globally fixed. Second, tokens located in upper regions can also drive mouth-related actions in the lower facial area, while tokens in lower regions may affect eyebrow movements in the upper facial area. **This indicates that tokens at a given 2D position do not consistently activate facial actions in the corresponding facial region, but are instead determined by their relative ordering and contextual interactions. Unlike CNNs that rely on local geometric adjacency to organize spatial semantics, our tokenizer composes motion representations through global attention across all sequence locations along with a globally shared codebook. Consequently, even global facial actions such as head movements can be controlled by manipulating a single token at a specific sequence location, rather than requiring the simultaneous activation of all tokens across the 2D grid** (see Appendix A.2 for more detailed motion composition studies and visualization results).”
>
> **For the choice of hyperparameters** in the C-VQ-VAE, we found that the **codebook size** and **latent action length** have the most significant impact on the final MER performance. Therefore, we conducted ablation studies to determine their optimal settings. The results are shown in Table 5. Thanks to large-scale facial video pretraining, the C-VQ-VAE can be trained stably and consistently converges to robust action representations.
>
> We have included a new Appendix A.4 to evaluate the stability of the Transformer for MER. Specifically, we conducted a sensitivity analysis on two key hyperparameters: the number of attention layers $L_{attn}$ and the hidden dimensionality $d$. Specifically, we evaluated models with {1, 2, 3, 4} layers and hidden dimensionality {256, 512, 768, 1024} while keeping all other settings unchanged.
>
> | $L_{attn}$ | $d$    | Composite (UF1) | Composite (UAR) |
> |--------|------|-----------------|-----------------|
> | 1      | 512  | 0.8789          | 0.8927          |
> | 2      | 512  | **0.8943**      | 0.8967          |
> | 3      | 512  | 0.8905          | **0.9032**      |
> | 4      | 512  | 0.8816          | 0.8967          |
> |--------|------|-----------------|-----------------|
> | 2      | 256  | 0.8764          | 0.8877          |
> | 2      | 512  | **0.8943**      | 0.8967          |
> | 2      | 768  | 0.8798          | **0.8990**      |
> | 2      | 1024 | 0.8717          | 0.8734          |
>
> As shown in Table 9 in Appendix A.3, the overall performance on the composite dataset under the CDE protocol dataset show that the overall performance varies only slightly across different settings, and all configurations achieve comparable accuracy. The model with 2 layers and a hidden size of 512 achieves a good trade-off between efficiency and performance, which we therefore adopt in the main experiments. These observations indicate that our results are **insensitive to reasonable hyperparameter changes**. These results also demonstrate that our framework is **a stable and reproducible** pipeline.
> In addition, we commit to releasing the source code to ensure reproducibility.

---

> ### Author Response · Authors · 2025-11-20
> **Response to Reviewer UyN7 (part 3)**
>
> **Weakness 4: The reported accuracy improvements are marginal (e.g., small gains over LTR-3O and SRMCL in Table 2), raising questions about practical significance. There is little ablation or visualization proving that the semantic tokens truly capture meaningful, interpretable muscle movements.**
>
> **A:** We acknowledge that the overall improvements on the composite dataset (Table 2) over previous methods such as LTR-3O and SRMCL may appear marginal. However, MER is a highly challenging task due to the subtle facial motions and limited data scale, where even 1–2% improvements are considered meaningful. In particular, our method achieves consistent gains on CASME-II and SMIC-HS datasets. In addition, we gain a notable +4.02% UF1 and +3.64% UAR improvement on the more challenging large-scale CAS(ME)3 dataset. Please note that the current SOTA method MER-CLIP leverages additional Action Unit annotations, while our method does not, which demonstrates the effectiveness of our framework.
>
> More importantly, beyond quantitative improvements, our framework introduces a new paradigm that transitions pixel-level motion to semantic-level action representation, offering interpretability and potential applications to ME generation, which can alleviate the current data scarcity issue in MER research.
>
> These results demonstrate both the quantitative and conceptual significance of our framework.
>
> For the second point, we have conducted extensive manipulation studies and analyses on the codebook and action tokens to verify that the tokens are identity- and domain-invariant. In addition, the results prove the interpretability of the tokens. Results are shown in Figure 3, Figure 4, Appendix A.1, and Appendix A.2. In the revised manuscript, we **also conducted additional experiments on in-the-wild facial images to further validate the effectiveness of the tokens**. In addition, we have added experiments on in-the-wild facial data in the revised version, where facial muscle locations and illumination are highly unconstrained. Our method still produces consistent and robust motion representations, which further validates our claim of domain and identity invariance through discretization. Please kindly refer to the **new Appendix A.6 and Figure 10**.
>
> **Question 1: How generalizable is D-FACE to in-the-wild, low-quality videos, given its dependence on large-scale pretraining on high-quality data?**
>
> **A:** To evaluate the cross-domain generalization ability of D-FACE, we **conducted qualitative experiments on a collection of in-the-wild facial images** obtained from unconstrained online sources. For each facial image, we first generated the token map that corresponds to the ``no-action'' state, and then selectively altered token indices to generate new facial images.
>
> To further evaluate robustness under low-quality inputs, we **downsampled each facial image to 20% of its original resolution** and then feed the degraded images into the facial action tokenizer. We repeated the same procedure: obtaining the “no-action” token map and applying the same token manipulations for facial image generation.
>
> As shown in Figure 10, D-FACE generalizes well on both high-resolution and low-quality in-the-wild images, generating natural faces with diverse facial actions. These results demonstrated that the learned facial action tokens encode stable and meaningful motion semantics that transfer reliably across identities, domains, and even under substantial resolution degradation.
>
> Please kindly refer to the **new Appendix 6 and Figure 10**.

---

> ### Author Response · Authors · 2025-11-20
> **Response to Reviewer UyN7 (part 4)**
>
> **Question 2: Could the discretization bottleneck of VQ-VAE lose fine-grained motion nuances crucial for micro-expression detection?**
>
> **A:** We acknowledge that the discretization in VQ-VAE may theoretically introduce some loss of fine-grained motion details. However, the quantized representations are much cleaner and capture only motion-related components, effectively filtering out irrelevant appearance variations. This property is particularly beneficial for micro-expression recognition, where the discriminative cues lie in facial motions rather than pixel-level reconstruction fidelity.
>
> To validate this, we conducted an additional experiment using the same pretrained C-VQ-VAE, where we still compute the feature difference between two input frames as before, but remove the subsequent discretization step, allowing the resulting motion features to remain in a continuous latent space. As shown in the table below, removing the discretization step leads to a clear drop in MER performance, even though the discretization may introduce some loss of reconstruction details. We have included this analysis in the new Appendix A.9 and Table 13. Please kindly refer to it.
>
> | Method                    | Composite (UF1) | Composite (UAR) |
> |--------------------------|-----------------|-----------------|
> | w/o discretization       | 0.8762          | 0.8799          |
> | w/ discretization (ours) | **0.8943**      | **0.8967**      |

---

### Official Review · Reviewer_ww2V · 2025-10-26

**Soundness:** 2
**Presentation:** 2
**Contribution:** 2
**Rating:** 4
**Confidence:** 3

**Summary:**

This work is about micro expression recognition. MER is very useful for many applications yet very difficult to detect. This work set the task in a semantic framework, instead of relying on pixel-level MER, as most existing work did. A Transformer was used to process flattened latent feature vector, followed by linear projection and alignment to identify micro expression. Results were good and ablation study was supportive of the necessity of each step.

**Strengths:**

Strengths include the approach to shift from pixel-wise MER to semantics-based MER, and supportive ablation study.
It sounds like a reasonable idea to discretize the difference between two consecutive frames into identity- and domain-invariant action representations, but the ensuing description of the method did not seem to pursue this idea.

**Weaknesses:**

The writing is not always easy to follow. It is not clear why authors performed face generation by inserting facial token into various locations at the action token matrix. What was the purpose of this step? Is it just to demonstrate that locations of the same token leads to different facial expression?
In Eq. 5, what was the linear projection P? Can the authors give the detail?
A big concern is the impact of the size of codebook on the final performance, Table 5. It is puzzling that the size of the codebook would have such a big effect on the capacity, and it is worrisome that a larger codebook size actually led to reduced capacity, according to the table.
In the method, which part is the identity-invariant action representation and which part is the domain-invariant action representation? It seems these two representation were not described in the algorithm.

**Questions:**

Please see weaknesses.

---

> ### Author Response · Authors · 2025-11-20
> **Response to Reviewer ww2V (part 1)**
>
> Thank you very much for your constructive comments. Please find our responses to each of your concerns below.
>
> **Weakness 1: The writing is not always easy to follow. It is not clear why authors performed face generation by inserting facial token into various locations at the action token matrix. What was the purpose of this step? Is it just to demonstrate that locations of the same token lead to different facial expression?**
>
> **A:** The purpose of “performing face generation by inserting facial tokens into various locations” is **to investigate the semantics of the learned tokens and how they interact, which is necessary for correctly modeling them in the subsequent MER task**. After pretraining with the C-VQ-VAE objective, the facial action tokenizer is robust enough to generate an action token map from a facial image pair. However, this token map is produced in a 4×4 matrix, and it is unclear how individual tokens function within this structure. Therefore, we conducted extensive manipulation studies by performing “face generation by inserting facial token into various locations at the action token matrix”. Through the manipulation studies shown in Figure 3, we yield two key observations:
>
> 1)	The same action token may activate different facial muscle motions when placed at different sequence locations, indicating that **a token’s semantics depend on its sequence location** rather than being globally fixed.
>
> 2)	Tokens located in the upper facial region may also drive mouth-related actions, while tokens in the lower region may affect eyebrow movements. This suggests that **the tokens at a given 2D position do not consistently activate facial actions in the corresponding facial region.**
>
> In summary: The **order** of the token sequence is important, while **2D position** is irrelevant.
>
> In addition to the examples shown in Figure 3, we also conducted extensive manipulation studies to explore how each token works. Through extensive manipulation studies, we observed that each token at a fixed sequence location corresponds to consistent motion semantics (see Table 8 in Appendix A.2). **Tokens at different sequence locations can further interact through the global attention mechanism** to compose more complex motions or modulate the motion intensity (see Figure 8 in Appendix A.2).
>
> Therefore, unlike CNNs that rely on local geometric adjacency to organize spatial semantics, our tokenizer determines motion composition through global attention across all sequence locations along with a globally shared codebook. As a result, even global facial actions such as head movements can be generated by manipulating a single token at a specific sequence location, without requiring simultaneous activation of all tokens across the 2D coordinate grid ((see new Figure 3(d))). **These observations and analyses naturally motivate us to model action tokens as 1D sequences using a Transformer**. We have included these analyses in Section 3.2 Line 266–274. Please kindly refer to it.
>
> **Weakness 2: In Eq. 5, what was the linear projection P?**
>
> **A:** We would like to clarify that the linear projection P is a linear layer that maps the discrete token embeddings back to the high-dimensional feature space required by the decoder for image reconstruction. **We explicitly denote P in Section 3.1 because it is reused in the subsequent MER stage. Specifically, after obtaining the action tokens by feeding the onset and apex frames to the facial action tokenizer, we process these tokens with P to produce their high-dimensional representations, which serve as input to the Transformer.** (see Section 3.3 Line 278–280 for the re-usage of P.)

---

> ### Author Response · Authors · 2025-11-20
> **Response to Reviewer ww2V (part 2)**
>
> **Weakness 3: A big concern is the impact of the size of codebook on the final performance, Table 5. It is puzzling that the size of the codebook would have such a big effect on the capacity, and it is worrisome that a larger codebook size actually led to reduced capacity, according to the table.**
>
> **A:** Thank you for this insightful question. As shown in Table 5, the codebook size affects the semantic granularity of the learned representations, thereby influencing MER performance. Before conducting the experiments, we expected that a smaller codebook might lack sufficient representational capacity, while an increasingly larger codebook would lead to more stable and expressive motion semantics. However, the results show that **an overly large codebook destabilizes training and harms MER performance**. We attribute this to codebook redundancy, where many feature vectors in the codebook become **infrequently used or capture unnecessary details** rather than essential facial muscle motions, resulting in less consistent representations. Setting codebook size to 32 provides a good balance between representational capacity and stability, yielding semantically meaningful and discriminative motion tokens.
>
> As shown in our codebook analysis in Appendices A.1 and A.2, when the codebook size is set to 32, several tokens already exhibit high semantic similarity, differing mainly in feature magnitude (which controls facial motion intensity). Therefore, **further increasing the codebook size only introduces redundant feature vectors** that represent unnecessary details. We have included this analysis in Section 4.4 Line 447–453 of the revised manuscript. Please kindly refer to it.
>
> **Weakness 4: In the method, which part is the identity-invariant action representation and which part is the domain-invariant action representation? It seems these two representations were not described in the algorithm.**
>
> **A:** The **action tokens** generated from the pretrained C-VQ-VAE **are inherently identity- and domain-invariant action representation**. This is thanks to the VQ-VAE training objective and the scale of the pretraining data, which let the codebook learn identity- and domain-invariant action representations.
>
> The quantization step in the C-VQ-VAE introduces **a strong information bottleneck that forces the continuous motion features**, obtained by differencing feature representations of diverse face pairs, **to be mapped into a limited set of shared codebook vectors**. Since our large-scale pretraining data contains **over one million face pairs from more than 7,000 identities** with substantial variations in appearance and head pose, **a limited codebook cannot encode individual-specific morphological differences** (e.g., how “opening mouth” manifests in a specific individual’s mouth shape). These identity and appearance variations lie in a much larger and more complex subspace than the low-dimensional discrete codebook space. As a result, the optimization objective naturally encourages the quantization process to **capture stable and commonly observed motion patterns shared across identities while filtering out identity- or appearance-specific variations**, enabling the discretization to produce identity- and domain-invariant motion representations. In the revised manuscript, we have included this theoretical analysis at the end of Section 3.1 Line 234–244. Please kindly refer to it.
>
> The distributional analysis of codebook vectors in Appendix A.1, the manipulation studies of token semantics across different sequence locations in Appendix A.2, and the qualitative visualizations support our interpretation. In the revised manuscript, we further include additional qualitative experiments on in-the-wild facial images, where facial muscle locations, illumination, and identity characteristics are highly unconstrained. As shown in the new Appendix A.6 and Figure 10, our method still produces consistent and robust motion representations, which further validates our claim of domain and identity invariance through discretization.

---

### Official Review · Reviewer_hr6P · 2025-10-30

**Soundness:** 3
**Presentation:** 3
**Contribution:** 3
**Rating:** 8
**Confidence:** 2

**Summary:**

The authors are the first to employ semantic-level facial action tokens for MER and model these tokens as 1D sequences analogous to human language. Furthermore, they design an alignment method to establish explicit connections between action tokens and human-understandable emotions. Experiments demonstrate state-of-the-art accuracy and strong cross-subject and cross-domain generalization.

**Strengths:**

- The authors introduce a new perspective in the MER field: they argue that previous works operate in the pixel domain, making them highly sensitive to identity variations, and thus shift the focus toward a more robust feature domain.

- The authors experimentally demonstrate why 1D token sequences better represent expressions than 2D token maps with absolute positional information, thereby reinforcing the rationality of their design.

- Their ablation studies on the core components clearly verify the independent contributions of each module, including the Transformer, sparse pooling, and EDCLIP loss.

**Weaknesses:**

- Lacks ablation comparisons with different combinations of loss function weights.

- Shows sub-optimal performance on grayscale datasets.

**Questions:**

- The experimental section lacks an ablation study on the weighting of loss components. Would exploring different combinations of loss weights further improve the performance?

- The authors mention that the sub-optimal performance on SAMM is due to training on RGB images. Would training on grayscale images reduce the model’s reliance on color and improve performance?

---

> ### Author Response · Authors · 2025-11-20
> **Response to Reviewer hr6P**
>
> Thank you very much for your positive evaluation. Please find our responses to each of your concerns below.
>
> **Weakness and Question 1: The experimental section lacks an ablation study on the weighting of loss components. Would exploring different combinations of loss weights further improve the performance?**
>
> **A:** Thank you for this valuable suggestion. We indeed explored different loss weights during the development of our framework, although the results were not included in the main manuscript due to space limitations. Following the reviewer’s comment, we conducted an ablation study on the weights of the EDCLIP contrastive loss and the margin-based constraint loss for the class “others”. The results are provided in the new Appendix A.8 and Table 12.
>
> Results on the CASME-II dataset (5-class)
> | λ_EDCLIP | Acc     | UF1     | UAR     |
> |----------|---------|---------|---------|
> | 0.0      | 0.8394  | 0.8286  | 0.8384  |
> | 0.1      | 0.8434  | 0.8409  | 0.8604  |
> | 0.3      | **0.8474** | **0.8571** | **0.8694** |
> | 0.5      | 0.8474  | 0.8455  | 0.8631  |
> | 0.7      | 0.8394  | 0.8353  | 0.8429  |
>
> | λ_oth | Acc     | UF1     | UAR     |
> |-------|---------|---------|---------|
> | 0.0   | 0.8474  | 0.8472  | 0.8643  |
> | 0.2   | 0.8474  | **0.8571** | **0.8694** |
> | 0.4   | **0.8514** | 0.8365  | 0.8533  |
> | 0.6   | 0.8434  | 0.8365  | 0.8438  |
>
> For EDCLIP loss, we find that a small weight leads to insufficient contrastive alignment, while an overly large weight affects the classification objective. For margin-based constraint, we observe that overly large values put too much emphasis on separating the “others” samples from all textual embeddings, thus leading to performance degradation. Based on these observations, we set the weight of EDCLIP to 0.3 and the weight of margin-based constraint to 0.2 for every experiment.
>
> **Weakness and Question 2: The authors mention that the sub-optimal performance on SAMM is due to training on RGB images. Would training on grayscale images reduce the model’s reliance on color and improve performance?**
>
> **A:** Thank you for your insightful suggestion. Based on your feedback, we re-pretrained the facial action tokenizer by **partially converting the RGB facial pairs to grayscale with a probability of 0.5**, and then fine-tuned it on the composite dataset under CDE protocol. However, due to the limited rebuttal time, we re-pretrained the model starting from the 300k-step checkpoint, which had been trained on RGB images. From this checkpoint, we continued pretraining for another 300k steps using mixed RGB–grayscale images.
>
> However, as shown in the table below, the performance degrades after introducing grayscale images into pretraining. We believe this is because **mixing RGB and grayscale inputs alters the feature distribution and causes the learned action-token semantics to drift, destabilizing the pretrained codebook**. Since VQ-VAE is highly sensitive to input-domain statistics, such domain mixing can break the motion representations learned from the original large-scale RGB data.
> Nevertheless, we believe that a domain adaptation method such as training on grayscale images is a promising direction for reducing the model’s reliance on color, and we will further explore it as future work.
>
>
> Results on the composite dataset under the CDE protocol
> | Domain | Full UF1 | Full UAR | CASME-II UF1 | CASME-II UAR | SMIC-HS UF1 | SMIC-HS UAR | SAMM UF1 | SAMM UAR |
> |--------|----------|----------|---------------|---------------|--------------|--------------|-----------|-----------|
> | Mixed  | 0.8726   | 0.8847   | 0.9500        | 0.9545        | 0.8186       | 0.8351       | 0.8331    | 0.8257    |
> | RGB    | 0.8943   | 0.8967   | 0.9738        | 0.9754        | 0.8422       | 0.8476       | 0.8716    | 0.8513    |

---

### Official Review · Reviewer_mHMN · 2025-11-01

**Soundness:** 3
**Presentation:** 3
**Contribution:** 3
**Rating:** 6
**Confidence:** 4

**Summary:**

This paper presents D-FACE, a novel framework for micro-expression recognition (MER). The work's primary contribution is a shift away from traditional pixel-level motion representations (like optical flow), which are sensitive to identity, toward a learned, discrete, semantic-level representation called "facial action tokens." The method uses a C-VQ-VAE pretrained on large-scale video data to "tokenize" facial movements. It then models these tokens as a 1D sequence using a Transformer with sparse attention, motivated by an empirical analysis of token semantics. Finally, it aligns these tokens with human-understandable emotions using a novel CLIP-based loss (EDCLIP), which includes a specific margin-based objective to handle the ambiguous "others" category. The method achieves state-of-the-art results and, more importantly, demonstrates impressive cross-subject and cross-domain generalization in qualitative generation experiments.

**Strengths:**

1. The central idea of discretizing facial motion into semantic tokens is a well-motivated and significant departure from standard pixel-based MER methods. The authors clearly identify the key weaknesses of prior work (identity sensitivity and lack of semantics) and propose a direct solution.
2. The cross-identity and cross-domain generation experiments in Figure 4 are powerful evidence for the paper's main claim. The ability to transfer an ME from one person to another by only transferring the learned tokens, while preserving the target identity, convincingly demonstrates that the D-FACE representation is successfully disentangled from identity.
3. The framework is well-designed and internally consistent. The empirical analysis in Section 3.2, which finds that tokens behave like 1D sequences, provides a clear justification for using a Transformer (a choice validated in the ablations). The sparse attention mechanism is well-suited for the local nature of MEs, and the EDCLIP module's specific handling of the "others" category is an intelligent solution to a practical problem in emotion recognition datasets.

**Weaknesses:**

1. The leap from the empirical analysis (Section 3.2) to the 1D Transformer modeling is not fully convincing. The paper states that "token semantics are position-dependent" (e.g., token '2' means something different at position (3,3) than at (1,0) in Figure 3). However, it also claims the "absolute 2D spatial positions... do not correspond to specific locations" and that this motivates flattening to a 1D sequence. This seems contradictory. The observation that meaning is position-dependent could equally (or perhaps more strongly) motivate the use of a 2D Transformer with 2D positional embeddings. The paper would be stronger if it provided a clearer explanation for why a 1D sequence is the correct interpretation of this finding.
2. The C-VQ-VAE tokenizer is pretrained on VoxCeleb, a dataset of unconstrained *macro-expressions* (e.g., people talking in interviews). The model is then fine-tuned and evaluated on *micro-expressions*, which are defined by their subtle, rapid, and involuntary nature. This is a significant domain gap. The authors do not discuss whether pretraining on large, voluntary motions biases the codebook and prevents it from learning the extremely subtle motions unique to MEs.
3. The paper states it adapts a C-VQ-VAE architecture from a robotics paper [1] and "refined the codebook design and latent action length". However, the specific details of these refinements are not provided. This makes the core tokenizer component, which is responsible for the paper's main contribution, difficult to understand and reproduce. And more related works should be discussed to support the adaptation. [2]
4. The action encoder architecture in Figure 2(b) includes a "Causal Transformer" that processes features from the two input frames. The term "causal" implies temporal masking, but it is not explained how this is applied between just two static frames ($I_1$ and $I_2$). This component's function and design are unclear.

[1] Seonghyeon Ye, et al. Latent action pretraining from videos. Proceedings of the International Conference on Learning Representations, 2025.

[2] Chen, Haodong, et al. "Finecliper: Multi-modal fine-grained clip for dynamic facial expression recognition with adapters." Proceedings of the 32nd ACM International Conference on Multimedia. 2024.

**Questions:**

What are the current inference costs, and are there any optimization options?

---

> ### Author Response · Authors · 2025-11-20
> **Response to Reviewer mHMN (part 1)**
>
> Thank you very much for your positive evaluation. Please find our responses to each of your concerns below.
>
> **Weakness 1: The leap from the empirical analysis (Section 3.2) to the 1D Transformer modeling is not fully convincing. The paper states that "token semantics are position-dependent" (e.g., token '2' means something different at position (3,3) than at (1,0) in Figure 3). However, it also claims the "absolute 2D spatial positions... do not correspond to specific locations" and that this motivates flattening to a 1D sequence. This seems contradictory. The observation that meaning is position-dependent could equally (or perhaps more strongly) motivate the use of a 2D Transformer with 2D positional embeddings. The paper would be stronger if it provided a clearer explanation for why a 1D sequence is the correct interpretation of this finding.**
>
> **A:** Thank you for your comment. Our logic in Section 3.2 is as follows.
>
> Through the manipulation studies, we found:
>
> 1)	The same action token may activate different facial muscle motions when placed at different sequence locations (see Figure 3, token “2” as an example), indicating that **a token’s semantics depend on its sequence location** rather than being globally fixed.
>
> 2)	If the 2D spatial position were important, we would expect tokens in **upper regions** (e.g., (0, 0)–(1, 3)) to **control upper facial muscles such as the eyebrows**, and tokens in **lower regions** (e.g., (2, 0)–(3, 3)) to **control lower facial muscles such as the lips and chin**. However, the results of manipulation studies contradict this assumption: token “2” at (1, 0) also activates lower-facial movements (lip corner puller), and token “2” at (3, 3) activates upper-facial movements (lid tightener). These results suggest that although token order is important, **the tokens at a given 2D position do not consistently activate facial actions in the corresponding facial region**. Therefore, we model the token map as a 1D ordered sequence instead of a 2D spatial grid.
>
> In summary: The **order** of the token sequence is important, while **2D position** is irrelevant.
>
> In addition to the examples shown in Figure 3, we also conducted extensive manipulation studies to explore how each token works. Through extensive manipulation studies, we observed that each token at a fixed sequence location corresponds to consistent motion semantics (see Table 8 in Appendix A.2). **Tokens at different sequence locations can further interact through the global attention mechanism** to compose more complex motions or modulate the motion intensity (see Figure 8 in Appendix A.2).
>
> Therefore, unlike CNNs that rely on local geometric adjacency to organize spatial semantics, our facial action tokenizer determines motion composition through **global attention across all sequence locations along with a globally shared codebook.** As a result, even global facial actions such as head movements can be generated by manipulating a single token at a specific sequence location (see new Figure 3(d)), without requiring simultaneous activation of all tokens across the 2D coordinate grid. We have included these analyses in Section 3.2 Line 265–274. Please kindly refer to it.
>
> Except for the qualitative and theoretical analyses, we also conducted experiments using a 2D Transformer with 2D positional embeddings. The additional results in the Table 6 show that the Transformer with 1D positional embeddings outperforms both CNNs and 2D Transformer, demonstrating the advantage of 1D sequential modeling over 2D spatial structures in capturing order-dependent token semantics.
>
> Results on CASME-II (5-class)
> | Category             | Method                        | Acc     | UF1     | UAR     |
> |----------------------|-------------------------------|---------|---------|---------|
> | Recognition Network  | CNN Backbone                  | 0.8313  | 0.8120  | 0.8204  |
> |                      | Transformer with 2D PE        | 0.8327  | 0.8164  | 0.8234  |
> |                      | Transformer with 1D PE (ours) | **0.8474** | **0.8571** | **0.8694** |

---

> ### Author Response · Authors · 2025-11-20
> **Response to Reviewer mHMN (part 2)**
>
> **Weakness 2: The C-VQ-VAE tokenizer is pretrained on VoxCeleb, a dataset of unconstrained macro-expressions (e.g., people talking in interviews). The model is then fine-tuned and evaluated on micro-expressions, which are defined by their subtle, rapid, and involuntary nature. This is a significant domain gap. The authors do not discuss whether pretraining on large, voluntary motions biases the codebook and prevents it from learning the extremely subtle motions unique to MEs.**
>
> **A:** Thank you for your insightful comment. **Originally, we attempted to pretrain the tokenizer on CAS(ME)3**, a large-scale micro-expression dataset containing 80 hours of video and over 8 million frames, including more than 1,000 manually annotated micro-expressions and 3,400 macro-expressions. **However, the pretrained results were unsatisfactory.** The codebook failed to learn robust action representations. We attribute this to the fact that most image pairs with an 8–15 frame gap exhibit extremely subtle motion, providing insufficient motion diversity for effective codebook learning.
>
> Therefore, we employed VoxCeleb for pretraining. Even though VoxCeleb mainly consists of interview videos with obvious facial expressions, **our sampling strategy, which randomly selects two frames with an 8–15 frame temporal gap** (corresponding to the typical ME duration), **naturally produces image pairs that capture both subtle and obvious motions**. This leads to a much richer and more diverse set of motion patterns, which substantially improves the robustness of the learned action tokens. In the revised manuscript, we show some sampled image pairs in the new **Appendix A.4 and Figure 9**. In addition, we commit to releasing the source code (including the data collection code for pretraining) and the pretrained model in the final version to ensure reproducibility.
>
> **Weakness 3: The paper states it adapts a C-VQ-VAE architecture from a robotics paper [1] and "refined the codebook design and latent action length". However, the specific details of these refinements are not provided. This makes the core tokenizer component, which is responsible for the paper's main contribution, difficult to understand and reproduce. And more related works should be discussed to support the adaptation. [2]**
>
> **A:** The original C-VQ-VAE architecture used in Ye et al. (2025) was designed for robotic control tasks, with a small codebook (size = 8) and a short latent action length (= 4). In their setup, the learned tokens are used as labels in a token-occurrence classification task (i.e., the model predicts whether each token appears, independently of its location). However, when we directly applied this framework to facial data, we found that such a configuration was far from sufficient. The generated images were of very poor quality and failed to capture meaningful facial motion. This is because, for robot control, only a few discrete and abstract actions (e.g., move forward, move backward, turn left) are required, whereas **facial motion analysis requires modeling subtle and complex local muscle movements**, which the original configuration cannot handle. Although we did not include results for codebook size = 8 and action length = 4, Table 5 shows that even with size = 16 and length = 9, the MER performance remains poor.
>
> We therefore systematically increased the codebook size and adjusted the latent action length, which led to significantly better reconstruction quality and more meaningful motion representations. Moreover, we observed that the original “token classification” strategy was not suitable for our domain, since our manipulation studies revealed that **token semantics are order-dependent, and that complex facial muscle movements are composed through global contextual interactions among tokens**. Consequently, we developed our unique framework that leverages motion-token representations for micro-expression analysis.
>
> Unlike [2], which achieves adaptation by introducing additional networks, we adapt the C-VQ-VAE to the facial micro-expression domain **through a data- and analysis-driven redesign of the entire representation pipeline**, supported by extensive empirical investigation. We have included the explanation in Section 3.1 Line 189–198. In addition, we commit to releasing the source code to ensure reproducibility.

---

> ### Author Response · Authors · 2025-11-20
> **Response to Reviewer mHMN (part 3)**
>
> **Weakness 4: The action encoder architecture in Figure 2(b) includes a "Causal Transformer" that processes features from the two input frames. The term "causal" implies temporal masking, but it is not explained how this is applied between just two static frames (I_1 and I_2). This component's function and design are unclear.**
>
> **A:** Although we feed only two static frames, they are temporally ordered: the earlier frame is I_1 and the later frame is I_2. Our objective is to learn action representations describing the transition from I_1 (onset frame) to I_2 (apex frame). Accordingly, we enforce a causal dependency: **I_2 may condition on I_1, but information must not flow from I_2 back to I_1**. We have revised the description about the causal Transformer to “The resulting features are subsequently passed through a causal Transformer to model **the directed temporal transition from I_1 to I_2**.” Please kindly refer to Line 201 in the revised manuscript.
>
> **Question 1: What are the current inference costs, and are there any optimization options?**
>
> **A:** We measured the inference cost of D-FACE on a 256×256 input pair using an NVIDIA A100 GPU. **The facial action tokenizer requires 0.118s per sample, and the Transformer-based classifier requires 0.002s per sample, resulting in a total inference time of roughly 0.12s per sample**. This speed is sufficient for offline MER scenarios.
>
> For optimization, since the downstream MER Transformer is already lightweight, the main computational cost lies in the facial action tokenizer. The components of the C-VQ-VAE can be further accelerated by replacing the encoder backbone with lightweight architectures and by distilling the tokenizer into a smaller model. We believe these directions can substantially reduce the inference cost, and we plan to explore them in future work.

---

### Official Review · Reviewer_MjG1 · 2025-11-07

**Soundness:** 3
**Presentation:** 3
**Contribution:** 3
**Rating:** 6
**Confidence:** 3

**Summary:**

This paper proposes D-FACE, a novel framework for micro-expression recognition (MER) that transitions from pixel-level motion descriptors (e.g., optical flow, frame difference) to semantic-level discrete facial action tokens. The approach employs a conditional VQ-VAE to encode facial motion between onset and apex frames into discrete tokens, modeled as sequences with a Transformer using sparse attention pooling. Moreover, an emotion-description-guided CLIP (EDCLIP) module aligns visual features with textual emotion semantics, improving interpretability and generalization. Extensive experiments on CASME-II, SMIC-HS, SAMM, and CAS(ME)3 demonstrate state-of-the-art performance and superior cross-domain generalization.

**Strengths:**

1. The paper convincingly reframes MER from a low-level motion estimation problem to a semantic-level action representation problem — an important conceptual contribution.

2. The architecture integrates C-VQ-VAE for discrete motion tokenization, Transformer-based sequential modeling, and EDCLIP for semantic alignment. Each component is well-motivated and supported by ablation studies.

3. Empirical thoroughness: Results on multiple MER benchmarks (CASME-II, SMIC-HS, SAMM, CAS(ME)3) are comprehensive. Ablation studies (Table 6) validate each design choice (Transformer, sparse pooling, CLIP alignment). Cross-identity and cross-domain generation experiments (Fig. 4) provide strong qualitative evidence of generalization.

4. The codebook visualization and token analysis (Appendix A.1–A.2) demonstrate that the learned discrete tokens correspond to interpretable, position-dependent facial actions — a rare property in MER models.

**Weaknesses:**

1. While empirically effective, the paper could strengthen its argument about why discretization (VQ-VAE) leads to domain invariance — possibly through statistical or theoretical analysis of identity disentanglement.

2. The large-scale pretraining dataset for the facial video tokenizer is only mentioned briefly. It’s unclear: Was identity information explicitly removed or balanced? How was pretraining validated before fine-tuning on MER datasets?

3. The contrastive objective (Eq. 12–13) is well-motivated, but comparisons with existing CLIP-based MER methods (e.g., MER-CLIP, 2025) could include a clearer ablation or embedding visualization to show how EDCLIP differs in practice.

4. The authors claim D-FACE enables generalizable facial action understanding, but experiments are limited to MER datasets. It would strengthen the paper to evaluate transferability to macro-expression or AU-based tasks.

5. On the SAMM dataset, performance drops due to grayscale input. The authors acknowledge this but could have mitigated it with color-agnostic pretraining or domain adaptation.

**Questions:**

What is the scale and composition of the “large-scale facial video data” used for D-FACE pretraining?

How does the choice of codebook size (K=32) affect semantic granularity? Could larger K provide richer action semantics, or would that harm stability?

Did you explore temporal modeling beyond onset–apex pairs (e.g., full ME sequences)?

How sensitive is EDCLIP performance to the choice of emotion text prompts?

---

> ### Author Response · Authors · 2025-11-20
> **Response to Reviewer MjG1 (part 1)**
>
> Thank you very much for your positive evaluation. Please find our responses to each of your concerns below.
>
> **Weakness 1: While empirically effective, the paper could strengthen its argument about why discretization (VQ-VAE) leads to domain invariance — possibly through statistical or theoretical analysis of identity disentanglement.**
>
> **A:** We thank the reviewer for this valuable suggestion. Although we do not have a strict theoretical analysis, we can provide a reasonable explanation. The quantization step in the C-VQ-VAE introduces a **strong information bottleneck that forces the continuous motion features**, obtained by differencing feature representations of diverse face pairs, **to be mapped into a limited set of shared codebook vectors**. Since our large-scale pretraining data contains **over one million face pairs from more than 7,000 identities** with substantial variations in appearance and head pose, a limited codebook **cannot encode individual-specific morphological differences** (e.g., how “opening mouth” manifests in a specific individual’s mouth shape). These identity and appearance variations lie in a much larger and more complex subspace than the low-dimensional discrete codebook space. As a result, the optimization objective naturally encourages the quantization process to **capture stable and commonly observed motion patterns shared across identities** while filtering out identity- or appearance-specific variations, enabling the discretization to produce identity- and domain-invariant motion representations. In the revised manuscript, we have included this analysis at the end of Section 3.1 Line 234–244. Please kindly refer to it.
>
> We have some statistical analyses of the codebook vectors. The distributional analysis of codebook vectors in Appendix A.1, the manipulation studies of token semantics across different sequence locations in Appendix A.2, and the qualitative visualizations support our interpretation. In the revised manuscript, we further include additional qualitative experiments on in-the-wild facial images, where facial muscle locations, background, and identity characteristics are highly unconstrained. As shown in the new Appendix A.6 and Figure 10, our method still produces consistent and robust motion representations, which further validates our claim of domain and identity invariance through discretization.
>
> **Weakness 2: The large-scale pretraining dataset for the facial video tokenizer is only mentioned briefly. It’s unclear: Was identity information explicitly removed or balanced? How was pretraining validated before fine-tuning on MER datasets?**
>
> **A:** Regarding the first question, we would like to clarify that we **did not explicitly remove or balance the identity information** during pretraining. Instead, we rely on the VQ-VAE objective and the large-scale pretraining dataset to let the codebook learn identity- and domain-invariant facial action representations. The detailed explanation can be found in our response to the previous question.
>
> For the second question, since we used the VoxCeleb dataset, which provides a validation subset, we monitored the reconstruction quality on this subset during pretraining. Specifically, we stopped pretraining when the reconstructed 𝐼_2 could accurately convey the facial motion consistent with the ground-truth 𝐼_2. Empirically, we found that the model achieved stable reconstruction behavior at around 600k–800k training steps, at which point pretraining was terminated before fine-tuning on MER datasets.
>
> Due to the space limitations, the section of pretraining details was in the Appendix in the original manuscript. Now we have moved it to Section 3.5 in the main manuscript and included more details. In addition, we show some sampled image pairs in the Appendix A.4 and Figure 9 in the revised manuscript. Please kindly refer to it.

---

> ### Author Response · Authors · 2025-11-20
> **Response to Reviewer MjG1 (part 2)**
>
> **Weakness 3: The contrastive objective (Eq. 12–13) is well-motivated, but comparisons with existing CLIP-based MER methods (e.g., MER-CLIP, 2025) could include a clearer ablation or embedding visualization to show how EDCLIP differs in practice.**
>
> **A:** Thank you for the insightful suggestion. We would like to clarify the difference in our motivation for introducing CLIP-based contrastive learning. **MER-CLIP** employs a video encoder to process the entire video and relies on Action Unit (AU)-based prompts to **regularize the video encoder toward learning action information**. In contrast, since **our facial action tokenizer has already learned appropriate action representations**, our contrastive objective focuses on **bridging the learned action information with human-understandable emotions**. Therefore, our EDCLIP is emotion-oriented and considers the “others” class, whereas MER-CLIP does not. This also explains why MER-CLIP relies on accurate AU annotations.
>
> However, AU labels are not always available in many datasets, especially in real-world applications. For example, the SMIC dataset does not provide AU annotations, so MER-CLIP cannot be evaluated on SMIC, while our method can. This demonstrates our better general applicability.
>
> Nevertheless, we also conducted an experiment by replacing the general action descriptions in our prompts with precise AU annotations following the AU description used in MER-CLIP. However, as shown in Table 10 of the new Appendix A.5, the MER performance degraded.
>
>
> Results on CASME-II (5-class)
> | Template   | ACC    | UF1    | UAR    |
> |------------|--------|--------|--------|
> | EDCLIP with AU annotations | 0.8273 | 0.8255 | 0.8231 |
> | Ours       |**0.8474** | **0.8571** | **0.8694** |
>
> In our original design, **all samples of the same emotion category share the same textual prompt** and therefore the same text embedding. This creates a stable semantic anchor for each emotion class, so different action-token sequences expressing the same emotion are consistently pulled toward the same point in the embedding space. This stabilizes the contrastive alignment.
> In contrast, AU-based descriptions vary across samples even within the same emotion class, since different ME instances of the same emotion may activate different AU combinations. As a result, samples of one emotion do not share a common text embedding but are aligned to different AU-based textual vectors. This disables the class-level semantic anchor in the text space, causing samples of the same emotion to be pulled in different directions, which makes the training less stable and reduces recognition performance. This result indicates that our action tokenizer has already learned robust and discriminative motion representations without the requirements of precise AU annotations.

---

> ### Author Response · Authors · 2025-11-20
> **Response to Reviewer MjG1 (part 3)**
>
> **Weakness 4: The authors claim D-FACE enables generalizable facial action understanding, but experiments are limited to MER datasets. It would strengthen the paper to evaluate transferability to macro-expression or AU-based tasks.**
>
> **A:** Thank you for your constructive suggestion. We also believe that our proposed framework has potential for many applications beyond MER. However, most macro-expression recognition methods do not rely on onset–apex frame pairs but instead rely on full video sequences or single static frames. This mismatch in input makes a direct comparison unfair and methodologically inconsistent. Therefore, we evaluate our framework on the **micro-expression action unit (AU) recognition** task. Although the AU annotations come from MER datasets, the AU task is independent of emotion recognition and **directly evaluates the effectiveness of the learned motion representations**. Specifically, we followed the evaluation protocol of [1] and conducted micro-expression AU recognition on the CASME-II dataset.
>
> For this experiment, we keep the setting exactly the same as in our MER framework:
> 1)	we use the same pretrained facial action tokenizer,
> 2)	we use the same Transformer architecture with the same hyperparameter settings,
> 3)	but we remove the EDCLIP module, as AU recognition does not require aligning the motion representation to textual emotion semantics.
>
> Action Unit Recognition Results on CASME-II.
>
> | Method          | AU1    | AU2    | AU4    | AU7    | AU12   | AU14   | AU15   | AU17   | Avg    |
> |-----------------|--------|--------|--------|--------|--------|--------|--------|--------|--------|
> | *Li et al.* [1] | 0.726  | 0.721  | **0.898** | 0.569  | **0.796** | **0.685** | 0.715  | 0.700  | 0.726  |
> | Ours            | **0.902** | **0.868** | 0.849  | **0.660** | 0.667  | 0.636  | **0.828** | **0.800** | **0.776** |
>
>
>
> Without any task-specific considerations and module designs, our framework achieves an average F1-score of 0.776 on the CASME-II dataset. We compare with the most recent work published in established venues. Despite not being designed for AU recognition, the proposed semantic-level facial action tokens remain robust and effective, demonstrating its transferability across tasks.
>
> We have included this part in Appendix A.7 in the revised manuscript. Further exploration in video sequence-based macro-expression recognition and other facial action-related applications, such as engagement estimation, will be part of our future work.
>
> [1] Li, Yante, Wei Peng, and Guoying Zhao. "Micro-expression action unit detection with dual-view attentive similarity-preserving knowledge distillation." Proceedings of the International Conference on Automatic Face and Gesture Recognition (FG 2021). IEEE, 2021.
>
> **Weakness 5: On the SAMM dataset, performance drops due to grayscale input. The authors acknowledge this but could have mitigated it with color-agnostic pretraining or domain adaptation.**
>
> **A:** Thank you for your insightful suggestion. Based on your feedback, we re-pretrained the facial action tokenizer by partially converting the RGB facial pairs to grayscale with a probability of 0.5, and then fine-tuned it on the composite dataset under CDE protocol. However, due to the limited rebuttal time, we re-pretrained the model starting from the 300k-step checkpoint, which had been trained on RGB images. From this checkpoint, we continued pretraining for another 300k steps using mixed RGB–grayscale images.
>
> However, as shown in the table below, the performance degrades after introducing grayscale images into pretraining. We believe this is because mixing RGB and grayscale inputs alters the feature distribution and causes the learned action-token semantics to drift, destabilizing the pretrained codebook. Since VQ-VAE is highly sensitive to input distributions, such domain mixing can break the motion representations learned from the original large-scale RGB data.
> Nevertheless, we agree that domain adaptation is a promising direction for improving cross-domain robustness, and we will further explore domain adaptation strategies as future work.
>
> Results on the composite dataset under the CDE protocol
> | Domain | Full UF1 | Full UAR | CASME-II UF1 | CASME-II UAR | SMIC-HS UF1 | SMIC-HS UAR | SAMM UF1 | SAMM UAR |
> |--------|----------|----------|---------------|---------------|--------------|--------------|-----------|-----------|
> | Mixed  | 0.8726   | 0.8847   | 0.9500        | 0.9545        | 0.8186       | 0.8351       | 0.8331    | 0.8257    |
> | RGB    | 0.8943   | 0.8967   | 0.9738        | 0.9754        | 0.8422       | 0.8476       | 0.8716    | 0.8513    |

---

> ### Author Response · Authors · 2025-11-20
> **Response to Reviewer MjG1 (part 4)**
>
> **Question 1: What is the scale and composition of the “large-scale facial video data” used for D-FACE pretraining?**
>
> **A:** We pretrain the C-VQ-VAE on the large-scale facial video dataset VoxCeleb. The VoxCeleb dataset consists of unconstrained interview videos collected from YouTube, where frame rates vary depending on the source video (most videos are around 25–30 fps). For pretraining, we randomly sample image pairs with a temporal gap between 8 and 15 frames, which approximately corresponds to the typical duration of micro-expressions (0.25–0.5 seconds). This sampling strategy yields **over one million face pairs from more than 7,000 distinct identities.**
>
> As we discussed in the response to the Weakness #2, we have moved this subsection back to the main manuscript Section 3.5, as we believe these details are essential in our method. In addition, we show some sampled image pairs in the Appendix A.4 and Figure 9 in the revised manuscript.
>
> **Question 2: How does the choice of codebook size (K=32) affect semantic granularity? Could larger K provide richer action semantics, or would that harm stability?**
>
> **A:** Thank you for this insightful question. As shown in Table 5, the codebook size indeed affects the semantic granularity of the learned representations, thereby influencing MER performance. Before conducting the experiments, we expected that a smaller codebook might lack sufficient representational capacity, while an increasingly larger codebook would lead to more stable and expressive motion semantics. However, the results show that **an overly large codebook destabilizes training and harms MER performance**. We attribute this to **codebook redundancy**, where many feature vectors in the codebook become **infrequently used or capture unnecessary details** rather than essential facial motions, resulting in less meaningful representations. K=32 provides a good balance between representational capacity and stability, yielding semantically meaningful and discriminative motion tokens. As shown in our codebook analysis in Appendices A.1 and A.2, when K=32, several tokens already exhibit high semantic similarity, differing mainly in feature magnitude (which controls facial motion intensity). Therefore, **further increasing the codebook size only introduces redundant feature vectors** that represent unnecessary details.
>
> **Question 3: Did you explore temporal modeling beyond onset–apex pairs (e.g., full ME sequences)?**
>
> **A:** Thank you for this insightful question. In the MER field, most existing methods rely on motion extraction between the onset and apex frames, since the difference between these two frames is typically the most discriminative. Following this common practice, we also perform action tokenization based on onset–apex pairs.
>
> However, we believe that extending the action tokenization to full ME sequences is a meaningful direction, as it could remove the dependence on apex frame annotations. We consider this an important and promising direction for our future work.
>
> **Question 4: How sensitive is EDCLIP performance to the choice of emotion text prompts?**
>
> **A:** Except for the comparison with the AU-based prompts following MER-CLIP discussed in our response to Weakness #3, we also evaluate another variant of our template: “A [emotion] face showing [general facial action descriptions].” As shown in Table 10 in Appendix A.5, the general-action-based variant yields performance close to our default template, indicating that EDCLIP is not overly sensitive to minor template changes and remains stable and effective.
>
> Results on CASME-II (5-class)
> | Template   | ACC    | UF1    | UAR    |
> |------------|--------|--------|--------|
> | Variant #1 | **0.8514** | 0.8565 | 0.8617 |
> | Ours       | 0.8474 | **0.8571** | **0.8694** |

---

### Author Response · Authors · 2025-11-20
**General Response**

We sincerely thank all the reviewers for their feedback and constructive suggestions. We provide detailed responses to the reviewers to address their specific concerns, and the corresponding revisions in the manuscript are highlighted in blue.

In addition, since pretraining details are important for our proposed framework, we move this subsection from the Appendix to Section 3.5 in the main manuscript.

---

### Author Response · Authors · 2025-12-01
**Summary Comment for the new AC**

We sincerely thank the Area Chair for handling our paper. We take this opportunity to briefly summarize our work, the main reviewer comments, and our responses.

---
## Paper Summary

This paper proposes D-FACE, a new framework for micro-expression recognition (MER) that, for the first time, shifts from relying on pixel-level motion descriptors (e.g., optical flow and frame difference) to modeling semantic-level facial action tokens. This is achieved by employing a conditional VQ-VAE to encode facial motion between onset and apex frames into discrete tokens. With carefully designed large-scale facial video pretraining and extensive token manipulation studies, we reveal that facial action tokens exhibit order-dependent semantics, motivating us to model them as a 1D sequence using a Transformer with sparse attention pooling. Moreover, we introduce an emotion-description-guided CLIP (EDCLIP) module to bridge the learned facial action tokens with human-understandable emotions, and specifically enforce that ambiguous “others” samples remain semantically distant from all emotion-specific textual embeddings rather than being aligned to an ill-defined description.

Extensive quantitative experiments on CASME-II, SMIC-HS, SAMM, and CAS(ME)^3 demonstrate state-of-the-art performance, while comprehensive qualitative results verify cross-identity and cross-domain generalization as well as potential for facial expression generation, indicating broader contribution beyond recognition.

---
## Reviews and Responses

We received feedback from five reviewers. Overall, the reviewers acknowledged the motivation and conceptual novelty of our work. In particular, they appreciated our central idea of being the first to model micro-expressions using semantic-level facial action tokens, rather than relying on pixel-level motion cues, which have been a dominant paradigm in MER over much of the past decade.

Notably, even the more critical comments focused primarily on requests for clarification, analysis, and additional experiments, rather than fundamental concerns about the soundness or novelty.

In summary, the main reviewer comments and our responses are as follows (all changes have also been reflected in the revised manuscript):

**(a) Why VQ-VAE discretization enables identity- and domain-invariant facial action tokens**

We added more analyses based on the information bottleneck principle, clarifying how discrete quantization under large-scale identity-diverse pretraining **suppresses identity-specific variations** and **encourages the model to capture stable and shared facial motion patterns**. We also performed distributional and semantic analyses of the learned codebook vectors, together with qualitative evaluations on laboratory faces, in-the-wild faces, and low-resolution faces, showing that the learned tokens encode interpretable and domain-invariant facial motions.

This question also involves the pretraining details. The robust facial action tokens are enabled by our carefully designed large-scale facial video pretraining strategy. We have moved detailed pretraining descriptions from the appendix to the main manuscript, and commit to releasing the source code and pretrained model for reproducibility.

**(b) Codebook size effects and discretization risk**

We added additional analyses and clarified that an overly large codebook introduces redundancy and instability, while a codebook size of 32 offers an optimal balance.

Regarding the risk of information loss due to discretization, we provided further analysis and conducted an ablation study comparing discrete vs. continuous motion representations, showing that discretization significantly improves MER performance and validating its necessity.

**(c) Why 1D sequence modeling instead of 2D**

We clarified our logic and added further analyses for the choice of 1D sequence modeling. We also conducted a direct comparison with a 2D Transformer, showing that 1D modeling better captures order-dependent token semantics.

**(d) Choice of hyper-parameters and training stability**

We added ablation studies on the hyper-parameters (loss weights, Transformer encoder layers and hidden size), and analyzed the training stability.

**(e) Prompt sensitivity in EDCLIP**

We added a prompt ablation study showing EDCLIP is stable under minor template variations, while AU-based prompts (MER-CLIP, 2025) reduce performance by disrupting class-level semantic anchors.

**(f) Additional clarification and experimental analysis**

- The novel insights in the adaption of VQ-VAE and CLIP
- The use of a causal Transformer for two-frame input
- Additional application and future directions (e.g., micro-expression action unit recognition, domain adaption on grayscale dataset, and full ME sequence modeling)
- The caption of Figure 1(b)
- Inference cost

---
We believe the paper has been substantially strengthened. Thank you again for your time and consideration.

---

### Meta-Review · Area_Chair_5haB · 2025-12-23

**Summary:**

Reviewers agreed D-FACE is a meaningful step for MER: it replaces pixel-level motion with discrete semantic facial-action tokens (VQ-VAE), models them with a Transformer, and uses EDCLIP for emotion alignment. Strengths were the clear paradigm shift, strong results across MER benchmarks, and convincing generalization demos. Main concerns were clarity/reproducibility, justification of design choices, and SAMM grayscale performance. Overall, the rebuttal substantially improved clarity and added supporting analyses/ablations. Recommendation: Accept.

**Reviewer Concerns:**

Addressed: added/moved pretraining details; clarified tokenizer/“causal” module and figure text; provided ablations on codebook size, loss weights, prompt sensitivity; added evidence for 1D modeling; showed discretization helps vs continuous features; reported inference cost.

Still outstanding: grayscale SAMM remains weaker (domain adaptation left future work); broader validation beyond MER/full-sequence settings is future work.

**Reviewer Scores:**

hr6P (8): likely 8
mHMN (6): likely 6
MjG1 (6): likely 6
UyN7 (6): likely 6
ww2V (4): likely 5

---

### Decision · Program_Chairs · 2026-01-26

Accept (Poster)